# Proteolysis of histidine kinase VgrS inhibits its autophosphorylation and promotes osmostress resistance in *Xanthomonas campestris*

Chao-Ying Deng[1], Huan Zhang[1,2], Yao Wu[1], Li-Li Ding[1,2], Yue Pan[1,2], Shu-Tao Sun[3], Ya-Jun Li[1,4], Li Wang[1] & Wei Qian [1]

In bacterial cells, histidine kinases (HKs) are receptors that monitor environmental and intracellular stimuli. HKs and their cognate response regulators constitute two-component signalling systems (TCSs) that modulate cellular homeostasis through reversible protein phosphorylation. Here the authors show that the plant pathogen *Xanthomonas campestris* pv. *campestris* responds to osmostress conditions by regulating the activity of a HK (VgrS) via irreversible, proteolytic modification. This regulation is mediated by a periplasmic, PDZ-domain-containing protease (Prc) that cleaves the N-terminal sensor region of VgrS. Cleavage of VgrS inhibits its autokinase activity and regulates the ability of the cognate response regulator (VgrR) to bind promoters of downstream genes, thus promoting bacterial adaptation to osmostress.

[1] State Key Laboratory of Plant Genomics, Institute of Microbiology, Chinese Academy of Sciences, Beijing 100101, China. [2] College of Life Sciences, University of Chinese Academy of Sciences, Beijing 100049, China. [3] Department of Core Facility, Institute of Microbiology, Chinese Academy of Sciences, Beijing 100101, China. [4] The College of Forestry, Beijing Forestry University, Beijing 100083, China. These authors contributed equally: Chao-Ying Deng, Huan Zhang Correspondence and requests for materials should be addressed to W.Q. (email: qianw@im.ac.cn)

In both eukaryotic and prokaryotic cells, signalling pathways can be generally divided into two major modes according to the persistence of protein participants: reversible and irreversible. For example, proteolysis is an irreversible, post-translational signalling cascade that modulates cell physiology[1,2], whereas protein phosphorylation catalysed by receptor kinases is reversible[3]. Although, the regulatory mechanisms of these two kinds of cellular signalling pathways have been extensively investigated, the cross-regulation between them is not fully understood. Recent studies in eukaryotes revealed that the proteolysis of receptor kinases is critical for regulating development, apoptosis and tumour genesis[4–6]. However, in prokaryotic cells, how proteolysis modifies receptor kinases and controls their regulatory functions are incompletely understood.

In gram-negative bacteria, regulated proteolysis catalysed by PDZ-domain-containing proteases, such as the HtrA-family proteases (high temperature requirement A) and tail-specific proteases (Tsp), modulates multiple physiological pathways, including virulence, stress response, quorum-sensing, protein quality control and antibiotic resistance[7,8]. Thus, these proteases are potential molecular targets for the development of novel antibacterial agents[9]. These proteases contain multiple domains, including one or more PDZ domains, and they are mostly located in the periplasm of gram-negative bacteria[10]. PDZ domain-containing proteases regulate physiological processes by binding or cleaving their protein substrates in cells. For example, under stress conditions, the DegS protease of *Escherichia coli* cleaves anti-σ factor RseA to release σ^E from the membrane and then activates the transcription of various stress-response genes[11]. In *Vibrio cholerae*, a Tsp protease cleaves TcpP to initiate the subsequent proteolysis and inactivation of the membrane-bound transcription factor (TF)[12]. In addition to anti-σ factors and TFs, studies revealed that PDZ domain-containing proteases also process misfolded proteins[13], outer-membrane proteins, phosphatases[14], haemagglutinin[15] and autotransporters[16]. However, identification of the physiological substrates of proteases remains technically challenging, and thus, the regulatory functions of most proteases are unclear[17].

The majority of bacterial species belonging to the genus *Xanthomonas* are plant pathogens. Among them, *Xanthomonas oryzae* pv. *oryzae* is the causative agent of rice bacterial blight disease, and *Xanthomonas campestris* pv. *campestris* causes the black rot disease of crucifers. Both bacterial species encode at least six PDZ domain-containing proteases[18]. The inactivation of one orthologous gene, *prc* (also named *tsp*), results in the attenuation of virulence and hypersensitivities to multiple environmental stresses[18,19]. Prc of *X. oryzae* pv. *oryzae* is a periplasmic protein, and its cleavage substrate remains to be discovered. When the bacterium is grown under osmolarity stress (osmostress), Prc binds to a virulence-associated dipeptidyl peptidase (DppP). Rather than cleaving DppP, Prc stabilizes DppP, suggesting it has a chaperone activity[18]. Therefore, DppP is not a proteolytic substrate of Prc. More direct and effective approaches are necessary to identify the physiological substrates of this protease, which will allow the regulatory role of Prc to be determined.

In the present study, we aim at identify the physiological substrates of Prc and investigate the role of Prc-catalysed proteolysis in controlling the environmental adaptation of *X. campestris* pv. *campestris*, which is more genetically amenable than *X. oryzae* pv. *oryzae*. Under osmostress, Prc directly binds and cleaves the N-terminal sensor region of VgrS, a canonical histidine kinase (HK) that controls bacterial virulence and stress responses. Cleavage of the VgrS sensor inhibits its autokinase activity, which then modulates the TF activity of the cognate VgrR in binding the promoters of downstream genes. This process significantly promotes bacterial resistance to osmostress.

Our study reveal that the proteolytic processing of the sensor region of HK by protease is a molecular mechanism to control the TCS signalling of gram-negative bacteria.

## Results

**Prc regulates virulence and bacterial stress resistance.** In *X. campestris* pv. *campestris* 8004, XC_0714 is the orthologue of Prc (PXO_04290) in *X. oryzae* pv. *oryzae* PXO99^A (BlastP search, e-value = 0, identities = 94%). Both proteins encode three domains: an N-terminal PDZ domain, a central peptidase domain and a C-terminal DUF3340 domain with an unknown function (Fig. 1a). The Prc protein of *X. campestris* pv. *campestris* contains a predicted 26-aa signal peptide with a cleavage site between the 26th and 27th positions. According to a MEROPS peptidase database search result[20], Prc belongs to the S41A serine endopeptidase subfamily (e-value = $2.40 \times 10^{-99}$) that contains two conserved catalytic residues (Ser^475 and Lys^500). This is different from HtrA family proteases (S1C subfamily) that usually contain a catalytic triad (His–Asp–Ser). Western blotting revealed that the Prc of *X. campestris* pv. *campestris* localized to both the periplasm and cytosol (Fig. 1b). In addition, an RT-PCR analysis revealed that *prc* is located in a four-gene operon that also contains XC_0711, XC_0712, and XC_0713 because positive RT-PCR products were amplified from a cDNA template derived from the intergenic transcripts of these genes (Supplementary Fig. 1).

We constructed various *prc* mutants and characterized their phenotypic alterations, including biofilm development, virulence, the production of extracellular polysaccharides and enzymes, and resistance to multiple antibiotics and environmental stresses. In-frame deletions of the *prc* coding sequence or its PDZ (ΔPDZ), peptidase (ΔPEP) or DUF (ΔDUF) domains′ coding sequences independently caused significant decreases in bacterial virulence and *in planta* growth against susceptible cabbage (*Brassica oleraceae* cv. Jingfeng No. 1; Fig. 1c, d, Supplementary Fig. 2a–2c) and tolerance to the antibiotics erythromycin and kanamycin (Fig. 1e, f), as well as bacterial resistance to a high $Fe^{2+}$ concentration (2.5 mM; Fig. 1g) and osmostress (1.0 M sorbitol; Fig. 1h). Genetic complementation was performed by providing a full-length *prc* (Δprc-prc) or the heterogeneous *prc* of *X. oryzae* pv. *oryzae* PXO99^A (Δprc-prc_{Xoo}) and fully restored phenotypic deficiencies to levels similar to those of the wild-type (WT) strain (Fig. 1c–h, Supplementary Fig. 2). Thus, *prc* may regulate bacterial virulence and resistance to multiple environmental stresses, and the biological functions of the *prc* of *X. campestris* pv. *campestris* and *X. oryzae* pv. *oryzae* appear to be highly conserved. Here, we focused on the role of Prc in regulating bacterial responses to osmostress.

**Prc is a bona fide serine endopeptidase.** Our previous study failed to detect the in vitro peptidase activity of Prc in *X. oryzae* pv. *oryzae*[18]. In this work, a mature form of Prc from *X. campestris* pv. *campestris* was successfully obtained by expressing the full-length *prc* in the *Escherichia coli* BL21(DE3) strain. N-terminal amino acid sequencing of the purified, soluble Prc showed that the 26-aa signal peptide was removed, indicating that *E. coli* cells could recognize and process the cleavage site of the signal peptide. As shown in Fig. 2a and Supplementary Fig. 3a, the mature Prc of *X. campestris* pv. *campestris* not only degraded the universal substrate β-casein (Fig. 2a, Lanes 4 and 5) but also had endopeptidase activity that cleaved azocasein, a general chromogenic substrate for endopeptidases (Fig. 2b). When the core catalytic residues Ser^475 and Lys^500 were substituted, the degradative activities of recombinant Prc^S475A and Prc^K500A, respectively, were completely lost or significantly decreased (Fig. 2a, Lanes 8 and 9, and Lanes 12 and 13,

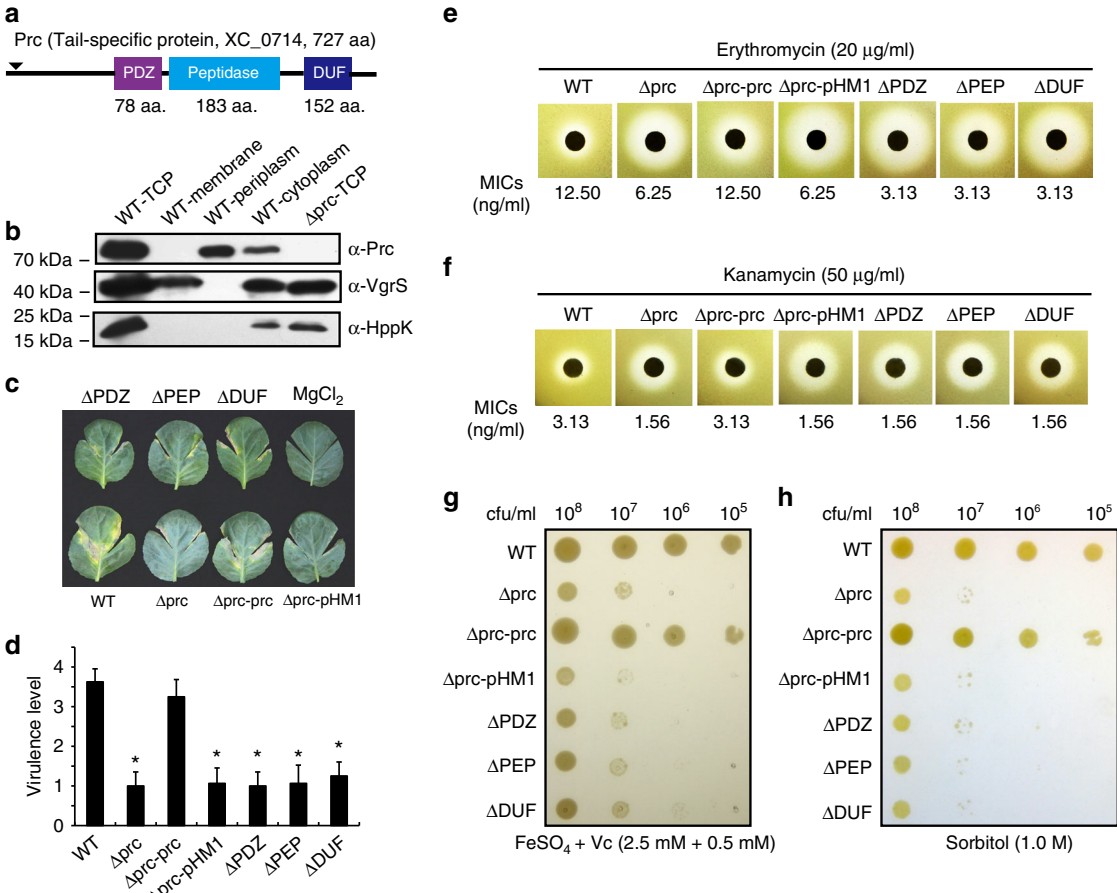

**Fig. 1** *prc* of *X. campestris* pv. *campestris* controls virulence and stress resistance. **a** Schematic view of the secondary structure of the Prc protein. Domain names are according to the pfam database. **b** Prc is located in bacterial periplasm and cytosol. Western blotting was used to detect Prc proteins in different cellular fractions. TCP total cellular protein. Western blotting of the membrane-bound and cytosolic proteins VgrS and HPPK, respectively, were used as controls. The experiment was repeated three times. **c** The inactivation of *prc* caused a deficiency in virulence. Bacterial strains were inoculated into leaves of the host plant *B. oleraceae* cv. Jingfeng No. 1. Virulence scores were estimated 10 d after inoculation. Sterile 10 mM MgCl$_2$ was used as the negative control. **d** Virulence scores of bacterial strains as shown in **c**. The virulence levels of bacterial strains were estimated using a semi-quantitative standard. Asterisks indicate significant differences relative to the WT strain (Student's *t*-test, $P < 0.05$, $n = 12$). The result of the *in planta* growth assay is shown in Supplementary Fig. 2. **e** and **f** The *prc* mutant is sensitive to various antibiotics, including erythromycin **e** and kanamycin **f**. In both **e** and **f**, the inhibitory zones of antibiotics are shown. The minimal inhibition concentrations (MICs) of the antibiotics were measured and are listed below. Each experiment was repeated three times. **g** The inactivation of *prc* resulted in hypersensitivity to iron stress. Bacterial strains were grown on NYG agar containing 2.5 mM FeSO$_4$ plus 0.5 mM vitamin C for 72 h at 28 °C. The experiment was repeated three times. **h** The inactivation of *prc* resulted in hypersensitivity to osmostress. Bacterial strains were grown on NYG agar containing 1.0 M sorbitol for 72 h at 28 °C. The experiment was repeated three times

respectively; Fig. 2b). Using Michaelis–Menten kinetic curve and Lineweaver–Burk plot analyses, Prc protease azocasein-hydrolysing activities were measured at 37 °C. As shown in Fig. 2c, d, the maximum protease activity and the Michaelis constant ($K_m$) value were 0.0023 ± 0.001 μmol/min/μg and 2.48 ± 0.32 μM, respectively. This kinetic value is similar to that of the Tsp from *E. coli* ($K_m$ = 4.4 ± 0.6 μM) and represents a relatively strong peptidase activity relative to other Tsp and HtrA proteases, which usually have $K_m$ values ranging from micromolar to nanomolar levels[21–23]. Prc protease activity was very stable under various conditions. The optimized temperature range for Prc activity was ~28–37 °C, and changing the pH value from 4.0 to 10.0 did not remarkably affect Prc activity (Supplementary Fig. 3b and 3c). In addition, when the dithiothreitol concentration in the reaction buffer was changed from 0 to 200 mM, the Prc activity level was only slightly affected (Supplementary Fig. 3d). These genetic and biochemical results confirmed the theoretical prediction that Prc is a serine endopeptidase.

**Monomeric Prc directly binds the VgrS sensor**. An affinity proteomic approach using tandem affinity purification (TAP) together with a nanoscale liquid chromatography–mass spectrometry (nanoLC–MS/MS) analysis[24] was employed to screen for Prc-binding proteins. Thus, a recombinant bacterial strain (Δprc-prc$^{S475A}$-HA-FLAG) was constructed for the TAP analysis. In this strain, HA and FLAG epitope tags were fused in tandem to the C-terminus of an inactive Prc in which the Ser$^{475}$ residue has been substituted with Ala. This substitution inactivated Prc protease activity to avoid the degradation of binding proteins during the analysis. The C-terminal HA and FLAG tags did not affect bacterial resistance to osmostress because the corresponding strain (Δprc-prc-HA-FLAG), which encoded a functional Prc, showed a growth rate similar to that of the WT strain on NYG–sorbitol plates (Supplementary Fig. 2f). The bacterial cells of the Δprc-prc$^{S475A}$-HA-FLAG strain were treated with osmostress consisting of 1.0 M sorbitol before being lysed by freeze grinding, which prevented the dissociation of Prc–protein

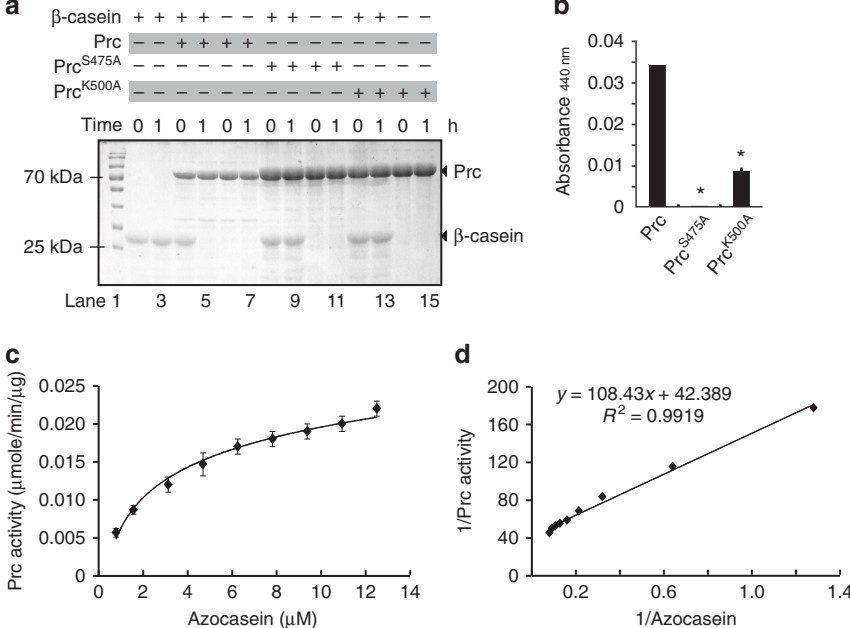

**Fig. 2** Prc is a serine endopeptidase. **a** Prc degrades β-casein, and its endopeptidase activity is dependent on the Ser[475] and Lys[500] sites. β-Casein (41.5 μM) was co-incubated with Prc (5 μM) or its recombinant forms (Prc$^{S475A}$ and PrcK$^{500A}$, 10 μM) at 28 °C for the indicated time. Reactions were stopped and analysed by SDS-PAGE together with Coomassie brilliant blue staining. **b** Quantification of Prc endopeptidase activity via degradation of the substrate azocasein. Azocasein (424 μM) was mixed with Prc (25 μM), and the reaction was carried out at 28 °C for 30 min. The optical absorbance was measured. Error bars represent standard deviations (n = 3). Asterisks indicate significant differences (Student's t-test, P < 0.05). **c** and **d** The Michaelis–Menten kinetics of Prc activity for azocasein hydrolysis. The Michaelis–Menten curve **c** and Lineweaver–Burk plot **d** were obtained from the specific reaction velocity of the hydrolysation of azocasein by Prc. The maximum specific $V_{max}$ and $K_{m}$ values of the Prc activity were determined from the graphic representations. The data were derived from three independent experiments, and the goodness of fit values ($R^2$) is indicated. **a–d**, the experiment was repeated three times

interactions as much as possible. With or without osmostress treatment, the affinity proteomic screening, through two independent rounds of immunoprecipitation, obtained 20 and 36 proteins, respectively, with 12 being present under both conditions (Fig. 3a). These 44 proteins were classified into 15 functional categories (Fig. 3b, Supplementary Table 3) and represent a subset of proteins potentially bound to Prc. Although further functional verification is needed, the diversity of these candidate proteins indicated that Prc may bind various proteins and modulate bacterial osmostress responses.

The TAP analysis identified a membrane-bound HK VgrS (XC_1050) that is important in regulating bacterial virulence and multiple-stress responses (Supplementary Table 3)[25]. VgrS is a metal receptor that contains a periplasmic, ferric iron-binding sensor, a transmembrane (TM) helix, a HAMP linker and a typical C-terminal transmitter domain (Fig. 3c, upper panel)[25,26]. Because both mature Prc and the sensor region of VgrS are located in the periplasm, we hypothesized that Prc bound to the VgrS sensor. As shown in Fig. 3c (lower left panel), a gel filtration analysis, together with analytical ultracentrifugation, revealed that Prc was present as a trimer (molecular weight = 272.43 ± 24.21 kDa) and monomer (molecular weight = 75.83 ± 10.17 kDa) in vitro. We purified a VgrS sensor peptide (Fig. 3c, lower right panel) and then used surface plasmon resonance (SPR) to measure the possible binding events between the VgrS sensor and different forms of Prc. The VgrS sensor physically bound the Prc monomer with a dissociation constant ($K_d$) of 33.9 μM (Fig. 3d), suggesting an intermediate level of protein–protein interaction. However, the VgrS sensor did not bind to the Prc trimer (Fig. 3e). A microscale thermophoresis assay (MST) also confirmed direct binding between the Prc monomer and VgrS sensor in solution ($K_d$ = 0.65 ± 0.090 μM; Supplementary Fig. 4a). Although Prc

bound the cytoplasmic region of VgrS that contained the HAMP and transmitter regions, the binding affinity was too weak to have any physiological significance ($K_d$ = 191 μM; Supplementary Fig. 4b and 4c). The difference in $K_d$ values detected by SPR and MST may be caused by the nature of these two approaches: SPR detects Prc–VgrS sensor interaction on a solid, CM5 sensor chip in which Prc is embedded, whereas the MST-associated binding event was directly measured in a solution, which is closer to the cellular environment.

To determine the region in Prc that binds the VgrS sensor, three recombinant Prc$^{S475A}$ proteins, each with a deletion of one of the domains (PDZ, PEP or DUF3340), were obtained and purified (Supplementary Fig. 4d). However, SPR analyses showed that none of these proteins bound the VgrS sensor (data not shown). A subsequent analysis revealed that the VgrS sensor bound a recombinant Prc$^{S475A}$ containing peptidase and DUF regions with a $K_d$ value of 22.4 μM (Fig. 3f). An MST analysis also confirmed that the VgrS sensor bound either the recombinant Prc$^{S475A}$ containing peptidase and DUF regions ($K_d$ = 0.27 ± 0.028 μM; Supplementary Fig. 4e) or another recombinant Prc$^{S475A}$ in which the PDZ domain was deleted ($K_d$ = 1.95 ± 0.113 μM; Supplementary Fig. 4f). Thus, the VgrS sensor bound the peptidase–DUF region of the Prc monomer rather than the PDZ domain that usually acts as a docking site in protein–protein interactions[27].

**Prc cleaves the VgrS sensor to inhibit autokinase activity.** VgrS is an HK with autokinase activity that is controlled by its periplasmic sensor region[25]. The Prc binding to the VgrS sensor region suggests that Prc regulates the latter's autophosphorylation. Thus, an in vitro phosphorylation assay was

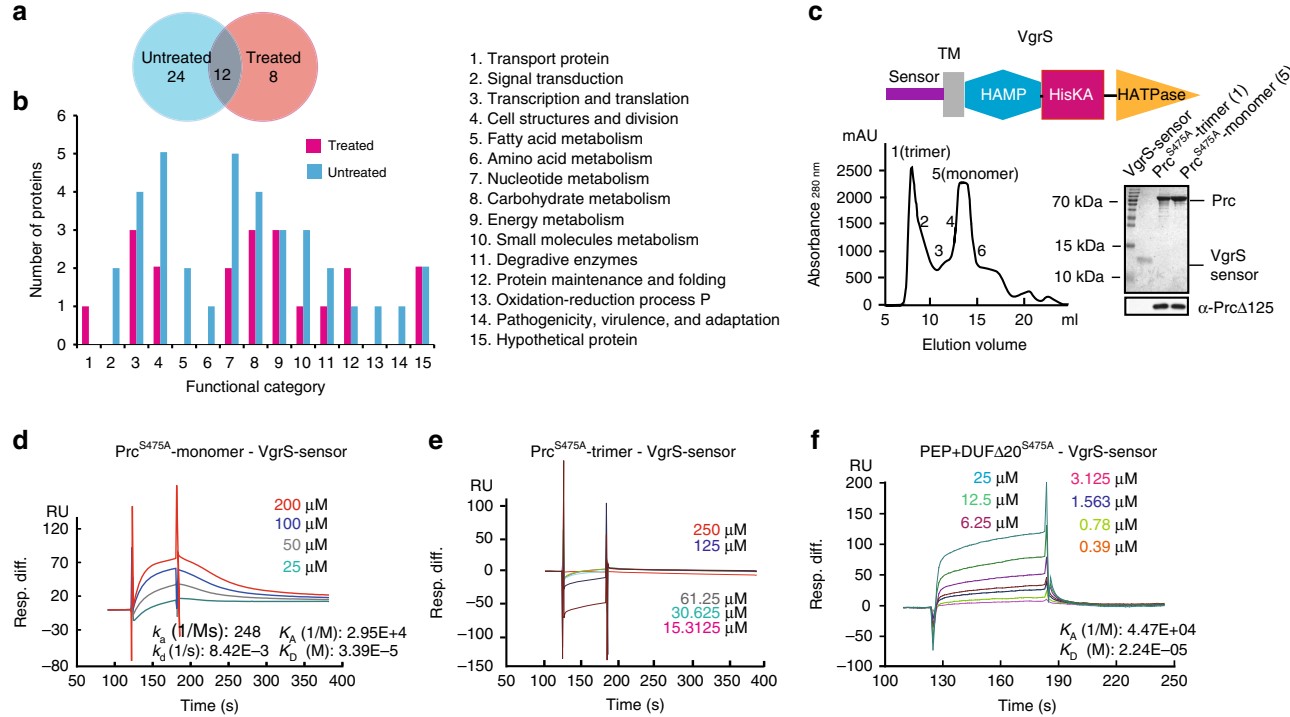

**Fig. 3** Tandem affinity purification (TAP) identify that the VgrS sensor directly binds the Prc monomer. **a** Venn diagram of the number of proteins identified by TAP together with a nanoLC−MS/MS analysis. Samples subjected or not subjected to osmostress were analysed. **b** Functional categories of the putative Prc binding proteins. Protein details are listed in Supplementary Table 3. **c** Prc exists as a monomer and a trimer in vitro. Upper panel: Secondary structure of VgrS. Lower left panel: Recombinant Prc$^{S475A}$ was separated by a molecular sieve, and the molecular weights of the fractions were measured by analytical ultracentrifugation. Lower right panel: Purification of the VgrS sensor and Prc proteins. Western blotting was used to verify the Prc proteins, as shown below. **d**–**f** Quantification of the binding affinity between the VgrS sensor and recombinant Prc by surface plasmon resonance. **d** The VgrS sensor bound the Prc$^{S475A}$ monomer. **e** The VgrS sensor did not bind the Prc$^{S475A}$ trimer. **f** The VgrS sensor bound a truncated Prc$^{S475A}$, which contains peptidase and DUF3340 domains. The VgrS sensor protein was trapped on a sensor CM5 chip, and various concentrations of Prc were injected at a flow-rate of 30 μl/min at 25 °C. Data were analysed using a model for a single set of identical binding sites. The binding kinetics of the Prc–VgrS sensor interaction: $k_a$ association rate constant; $k_d$ dissociation rate constant; $K_A$ equilibrium association rate constant; and $K_D$ equilibrium dissociation rate constant

performed to measure the VgrS autophosphorylation level in the presence of Prc. As shown in Fig. 4a, the addition of mature Prc to the reaction mixture decreased the autophosphorylation level of the full-length VgrS embedded in the inverted membrane vesicle (IMV) in less than 30 s (Lanes 5–8), while the addition of the inactive Prc$^{S475A}$ did not cause a similar effect (Lanes 9–12). This demonstrated that the protease activity of Prc is important in inhibiting the autophosphorylation of VgrS. In addition, neither mature Prc nor inactive Prc$^{S475A}$ affected the autokinase activity of the truncated, soluble VgrS that contained the transmitter domain (MBP-VgrS$^{cyto}$; Fig. 4b, Lanes 1–6 and Lanes 10–14, respectively), suggesting that Prc did not interact with the cytoplasmic region of VgrS. Because VgrS transfers the phosphoryl group onto the cognate response regulator (RR) VgrR[25], we also observed that the addition of Prc not only decreased the autophosphorylation of VgrS but also decreased the phosphorylation level of VgrR (Fig. 4c, Lane 6). Inactive Prc$^{S475A}$ did not have a similar impact (Fig. 4c, Lane 8). Thus, Prc may inhibit the autophosphorylation of full-length VgrS, and the sensor region of VgrS appears to be indispensable for inhibition.

To further investigate whether the VgrS sensor is a proteolytic substrate of Prc in vitro, the mature Prc protein was co-incubated with a purified, recombinant VgrS sensor. As shown in Fig. 4d, within 60 min, Prc cleaved the VgrS sensor into two parts and then completely degraded the sensor in vitro. However, when the inactive form of Prc (Prc$^{S475A}$) was added to the reaction mixture, the VgrS sensor remained intact during the co-incubation period

(Fig. 4e). Furthermore, to determine whether this proteolytic process took place in vivo, recombinant strains in the ΔvgrS mutant and the ΔprcΔvgrS double-mutant genetic backgrounds were constructed (ΔvgrS-vgrS$^{HA}$ and ΔprcΔvgrS-vgrS$^{HA}$, respectively; Supplementary Table 1). In the two strains, an HA-epitope tag was fused to the N-terminus of VgrS (between the 6th and 7th aa). Chloramphenicol was added to the bacterial cultures of the two strains to eliminate interference by de novo protein synthesis. Then, the bacterial cells were separately challenged with osmostress (1.0 M sorbitol). As revealed by western blotting analysis (Fig. 4f), after 5 min of osmostress stimulation, the amount of the N-terminal HA-tag, which was detected by the HA monoclonal antibody, gradually decreased as osmostress treatment progressed (Fig. 4f, upper panel, Lanes 6–10). Additionally, western blotting using a polyclonal antibody against VgrS revealed that VgrS was cleaved, and additional bands were detected (Fig. 4f, middle panel, Lanes 6–10). Thus, the N-terminal HA tag was removed from VgrS by Prc cleavage. In the double mutant, in which prc was deleted, the amount of HA-tag remained stable regardless of the presence or absence of osmostress stimulation (Fig. 4f, upper panel, Lanes 1–5), and no additional bands were detected by the polyclonal VgrS antibody (Fig. 4f, middle panel, Lanes 1–5). Collectively, the in vitro and in vivo experimental evidence demonstrated that the VgrS sensor is a proteolytic substrate of Prc. During the osmostress response, Prc directly bound and cleaved the VgrS sensor.

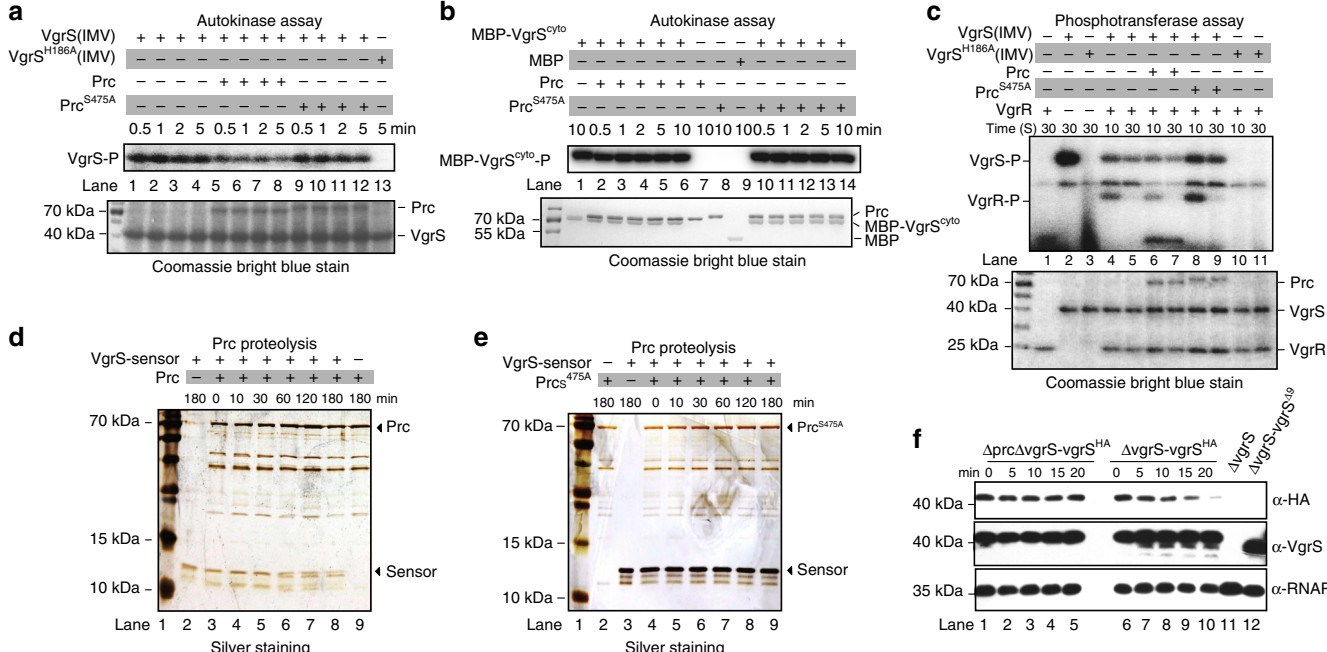

**Fig. 4** Prc cleaves the sensor region of VgrS to inhibit its autokinase activity. **a–c** Prc inhibited full-length VgrS autophosphorylation in a protease activity-dependent manner. **a** Full-length VgrS embedded in the inverted membrane vesicles (IMV, 10 μM) was phosphorylated by [γ-$^{32}$P]ATP. Before the addition of 10 μCi [γ-$^{32}$P]ATP, active Prc or inactive PrcS475A (2 μM) was added into the mixture. VgrS$^{H186A}$ IMV was used as a negative control of phosphorylation. **b** Prc did not affect the autophosphorylation of a soluble, cytosolic fragment of VgrS. In total, 10 μM soluble VgrS containing the transmitter region (MBP-VgrS$^{cyto}$) was used in the assay. **c** Prc inhibited the phosphorylation level of VgrR. After VgrS autophosphorylation, 10 μM VgrR was added into the mixture to elicit the VgrS–VgrR phosphotransfer reaction. In **a**, **b** and **c**: Upper panels show autophosphorylation assays; lower panels show Coomassie bright blue-stained gels used to check the amount of loaded protein. Aliquots were removed from the mixture at the indicated time points. The reaction was stopped by 6X SDS buffer, separated by SDS-PAGE and analysed by autoradiography. **d** and **e** Prc degraded the VgrS sensor in vitro. The purified VgrS sensor (100 μM) was co-incubated with 5 μM active Prc **d** or inactive Prc$^{S475A}$ **e** in enzymatic reaction buffer at 28 °C for the indicated time. Reactions were stopped, and the products were analysed by SDS-PAGE together with silver staining. **f** Western blotting revealed that Prc degraded N-terminal HA-tags in vivo. A bacterial strain that encoded recombinant VgrS fused with an HA-tag between the 6th and 7th residues was constructed independently in the ΔvgrS background. The bacterial strain was stimulated by 1.0 M sorbitol for different time periods. Total proteins were extracted, separated by SDS-PAGE, and analysed by western blotting. A polyclonal antibody of VgrS (α-VgrS) was used to measure the amount of VgrS protein, while monoclonal HA antibody (α-HA) was used to detect the N-terminal region of VgrS that was potentially cleaved by Prc, and the polyclonal antibody of RNAP (α-RNAP) was used as internal control. **a–f**, the experiments were repeated three times

To determine the exact, primary cleavage sites in the VgrS sensor processed by Prc, active Prc and VgrS sensor proteins were co-incubated, and then a matrix-assisted laser desorption/ionization time of flight mass spectrometry (MALDI–TOF–MS/MS) analysis was used to identify the VgrS sensor fragments. A peptide with a mass-to-charge ratio of 2771.5 was obtained (Fig. 5a, upper panel), and subsequent MS/MS analysis revealed that Prc cleaved the VgrS sensor at the site between Ala$^9$ and Gln$^{10}$ after 10 min of co-incubation (Fig. 5b, c, Supplementary Fig. 5). In the negative control experiment, the site remained intact when the inactive form of Prc$^{S475A}$ was added to the reaction mixture (Fig. 5a, middle panel). To verify this result in vitro, a recombinant VgrS sensor with the Ala$^9$ and Gln$^{10}$ sites substituted (Sensor$^{A9G-Q10A}$) was obtained and purified. MALDI–TOF–MS/MS analysis revealed that the Sensor$^{A9G-Q10A}$ protein was not cleaved by active Prc because no corresponding fragment was identified (Fig. 5a, lower panel). To verify this cleavage site in vivo, recombinant bacterial strains that encoded VgrS, in which the two sites were substituted in the ΔvgrS-vgrS$^{HA}$ and ΔprcΔvgrS-vgrS$^{HA}$ genetic backgrounds were constructed (ΔvgrS-vgrS$^{HA-A9G-Q10A}$ and ΔprcΔvgrS-vgrS$^{HA-A9G-Q10A}$, respectively; Supplementary Table 1). As shown in Fig. 5d, in the two recombinant strains, the N-terminal HA-tag of VgrS remained stable regardless of osmostress treatment (Fig. 5d, Lanes 1–5 and Lanes 6–10), indicating that recombinant VgrS$^{A9G-Q10A}$ resisted Prc cleavage

in vivo. As previously shown in Fig. 4a, b, after Prc cleavage, the difference in the theoretical molecular weights between the full-length and the truncated VgrS was ~1.0 kDa, and the two forms were hard to discriminate by SDS-PAGE analysis. Thus, we employed high-resolution, QTRAP LC–MS/MS analysis to directly detect the N-terminal proteolytic peptide generated by Prc cleavage. An 8-aa peptide (NRNIDFFA, corresponding to 2nd to 9th aa of VgrS) was chemically synthesized and used as a standard. As Fig. 5e and Supplementary Fig. 6 show, after a 5-min treatment of the full-length VgrS IMV by Prc, QTRAP LC–MS/MS analysis detected the NRNIDFFA peptide in the reaction mixture. The amounts of this peptide gradually increased along with the treatment time (Fig. 5e and Supplementary Fig. 6). These biochemical results demonstrated that Prc primarily cleaved the VgrS sensor between the Ala$^9$↓Gln$^{10}$ site in vivo and in vitro.

Furthermore, to measure the effect of Ala$^9$↓Gln$^{10}$ cleavage on the autokinase activity of VgrS, we obtained and purified a truncated, recombinant VgrS protein (VgrS$^{Δ9}$, embedded in IMV) containing a deletion from the 2nd to 9th aa at the N-terminus. An in vitro phosphorylation assay then revealed that the autophosphorylation level of the VgrS$^{Δ9}$ IMV was significantly decreased relative to that of the full-length VgrS IMV (Fig. 5f, Lane 2). As controls, two truncated VgrS IMV proteins (VgrS$^{Δ72}$ and VgrS$^{Δ84}$, which were truncated in the sensor region) retained phosphorylation levels that were similar or even

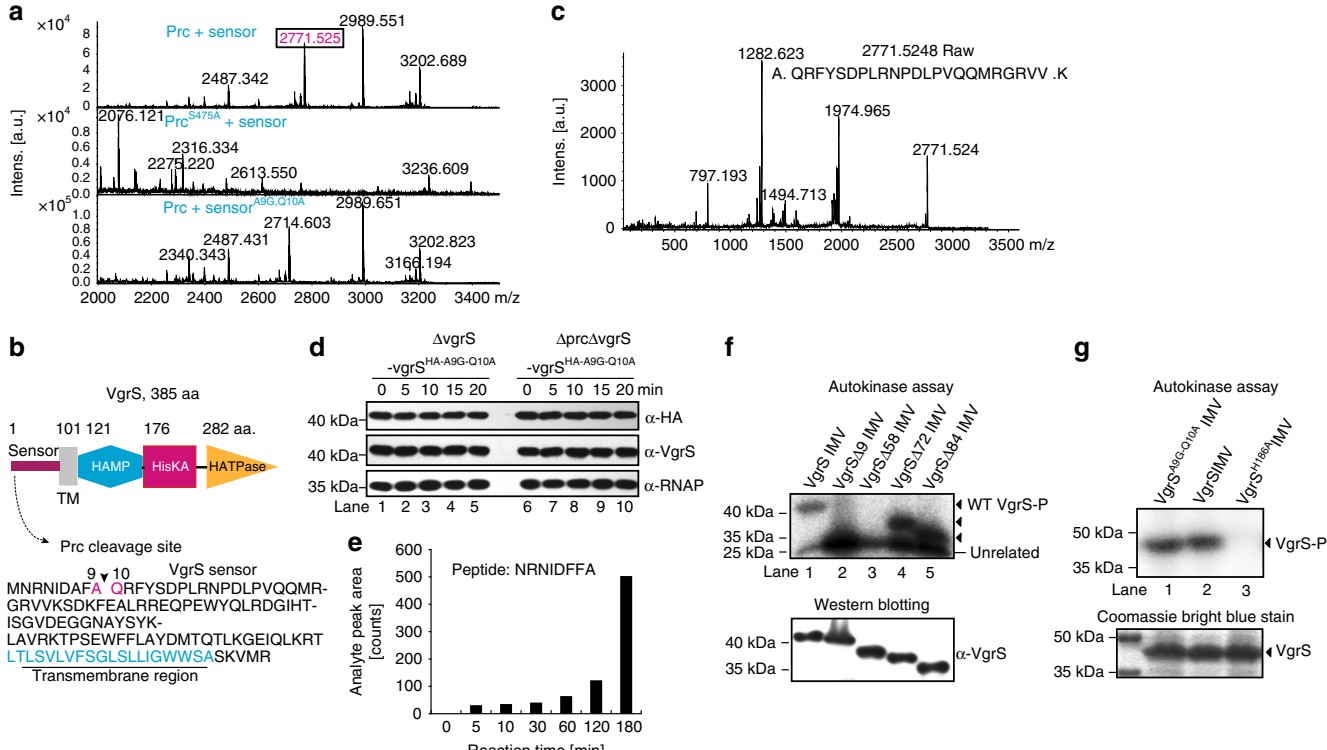

**Fig. 5** Identification of the VgrS cleavage site by the Prc protease. **a** MALDI–TOF–MS/MS analysis revealed that Prc cleaves the VgrS sensor at the Ala$^9$↓Gln$^{10}$ site. Upper panel: proteolysis of the VgrS sensor with active Prc. Middle panel: proteolysis of the VgrS sensor with inactive Prc$^{S475A}$, which was used as a negative control. Lower panel: proteolysis of the recombinant VgrS$^{A9G-Q10A}$ sensor with active Prc. Digested products were detected in positive ion reflectron mode over an $m/z$ range of 700–3500. Spectra showed relative intensities in the mass range of $m/z$ 2000–3400. **b** Schematic view of the secondary structure of VgrS and the cleavage site in the sensor region. TM transmembrane helix. **c** Identification of the Prc cleavage site in the VgrS sensor. A MALDI–TOF/TOF MS/MS spectral analysis of $m/z$ 2771.5 from a Prc + VgrS sensor sample. The magnified MS/MS spectra showed the fragment patterns of peptides. **d** Substitution of VgrS$^{A9G-Q10A}$ resulted in resistance to Prc cleavage. Bacterial strains that encoded recombinant VgrS$^{A9G-Q10A}$ fused with an HA-tag between the 6th and 7th residues were constructed independently in the ΔvgrS and ΔvgrSΔprc backgrounds. The bacterial strains were stimulated by 1.0 M sorbitol for different time periods. Total proteins were extracted, separated by SDS-PAGE, and analysed by western blotting. A polyclonal antibody of VgrS (α-VgrS) was used to measure the amount of VgrS protein, while a monoclonal HA antibody (α-HA) was used to detect the N-terminal region of VgrS that was potentially cleaved by Prc, and the polyclonal antibody of RNAP (α-RNAP) was used as an internal control. The experiment was repeated three times. **e** Detection of the N-terminal short peptide of VgrS generated by Prc cleavage. Full-length VgrS embedded in inverted membrane vesicles was treated by Prc, and the proteolytic products were analysed by QTRAP LC–MS/MS at different time points. A chemically synthesized peptide, NRNIDFFA, was used as standard. Details of the QTRAP LC–MS/MS analysis are shown in Supplementary Fig. 6. **f** and **g** Deletion of the N-terminal sequence of VgrS decreased its autophosphorylation level. **f** An in vitro phosphorylation assay was conducted to measure the autophosphorylation levels of the truncated VgrS$^{Δ9}$, VgrS$^{Δ58}$, VgrS$^{Δ72}$ and VgrS$^{Δ84}$ embedded in the IMVs (10 μM). The reaction was performed as described in Fig. 4a. **g** Substitution of VgrS$^{A9G-Q10A}$ did not affect its autokinase activity. Upper panels: autophosphorylation assay. Lower panels: western blotting of the proteins or Coomassie brilliant blue staining was used to check the amount of loaded protein. Two experimental repeats were performed

higher than that of the WT VgrS protein (Fig. 5f, Lanes 4 and 5), while the autokinase activity of VgrS$^{Δ58}$ was also substantially decreased (Fig. 5f, Lane 3). In addition, the recombinant VgrS$^{A9G-Q10A}$ protein had autokinase activity similar to that of the WT VgrS (Fig. 5g). Together with the data shown in Fig. 4a, the results of the phosphorylation analyses demonstrate that Ala$^9$↓Gln$^{10}$ of VgrS was the primary cleavage site of the Prc protease. Prc proteolysis resulted in a substantial decrease in the VgrS autophosphorylation level.

**Cleavage of VgrS promotes bacterial osmostress resistance.** The cleavage of a bacterial HK by a protease has not been previously reported, and the adaptive significance of the signalling process remains unclear. To genetically investigate the consequence of VgrS sensor cleavage on bacterial resistance to osmostress, an in-frame deletion mutant of vgrS (vgrS$^{Δ9}$), in which the coding sequence from the 2nd to 9th aa of the chromosomal vgrS was deleted (Fig. 5b), was constructed to mimic the truncated protein

product cleaved by Prc. Without osmostress, the mutant vgrS$^{Δ9}$ exhibited a growth rate similar to that of the WT strain (Fig. 6a). However, when this bacterial mutant was challenged by osmostress (1.0 M sorbitol), it grew faster than the WT strain (Fig. 6b). Under osmostress, genetic complementation, in which a plasmid-borne, full-length vgrS was provided to this vgrS$^{Δ9}$ mutant, decreased bacterial growth to the WT level (Fig. 6b). In addition, the growth of a bacterial strain encoding a recombinant VgrS$^{A9G-Q10A}$, which is resistant to Prc cleavage (vgrS$^{A9G-Q10A}$), was significantly reduced under stress conditions (Fig. 6d). Thus, cleavage of the VgrS sensor by Prc may be beneficial for the growth of *X. campestris* pv. *campestris* under osmostress.

Because Prc-catalysed VgrS cleavage resulted in a substantial decrease in its autokinase activity, we hypothesized that dephosphorylation of the VgrS–VgrR system promotes bacterial resistance to osmostress. As predicted, vgrS deletion mutants and point mutants of vgrS and vgrR, with substitutions in their conserved phosphorylating aa (vgrS$^{H186A}$ and vgrR$^{D51A}$,

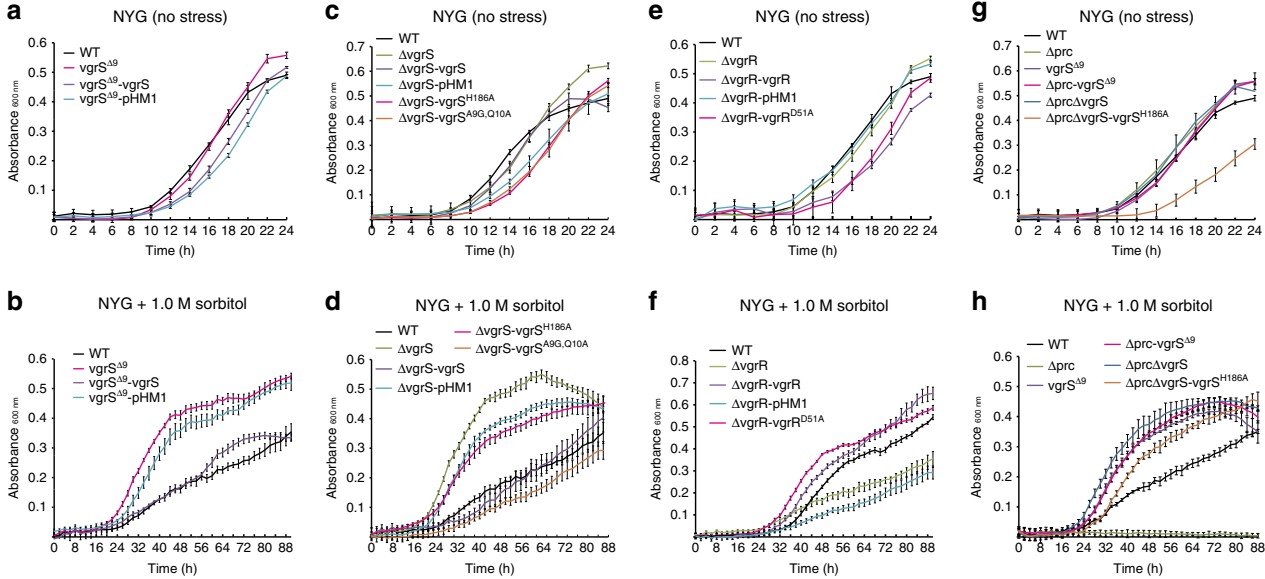

**Fig. 6** Deletion of the N-terminus of *vgrS* suppresses the growth deficiency of the *prc* mutant under osmostress. **a**, **c**, **e**, and **g** Bacterial growth in NYG medium. **b**, **d**, **f**, and **h** Bacterial growth in NYG medium plus 1.0 M sorbitol. The growth curves were measured by an automatic Bioscreen C instrument. Each data point is the average of six samples, and error bars indicate standard deviations

respectively), also exhibited remarkably increased capabilities to adapt to osmostress (Fig. 6c–f). However, similar to the *prc* mutant, the *vgrR* deletion mutant was highly susceptible to sorbitol treatment (Fig. 6e, f), suggesting that intact, dephosphorylated VgrR is indispensable for regulating bacterial responses to stress. To determine the epistatic relationship between *prc* and *vgrS*, a set of double mutants was constructed in the Δprc genetic background by mutating *vgrS*. As shown in Fig. 6g, h, the inactivation of *prc* almost completely arrested bacterial growth under osmostress, but the growth rates of the double mutants Δprc-vgrS$^{\Delta 9}$, ΔprcΔvgrS and ΔprcΔvgrS-vgrS$^{H186A}$ were significantly restored towards the WT level. These epistatic analyses demonstrate that mutations causing the dephosphorylation of VgrS efficiently suppressed the growth deficiency of the *prc* mutation in resisting osmostress. Thus, *vgrS* was located downstream of *prc* regulation during the bacterial osmostress response.

In addition to osmostress resistance, other phenotypic changes in the vgrS$^{\Delta 9}$ mutant were measured. The virulence of this mutant against a susceptible host cabbage (*B. oleraceae* cv. Jingfeng No. 1) decreased slightly, while the resistance of the vgrS$^{\Delta 9}$ mutant to iron stress and erythromycin was significantly decreased relative to the WT strain (Supplementary Fig. 7). Thus, under osmostress, the phosphorylation of VgrS was detrimental to bacterial growth. Prc-catalysed cleavage of the VgrS sensor to dephosphorylate this TCS system was specifically beneficial to bacterial osmostress adaptation.

**Prc controls the VgrR promoter-binding landscape**. VgrS autophosphorylates and then transfers the phosphoryl group to the cognate VgrR to control its TF activity in binding promoters[25]. Because Prc cleaved the VgrS sensor and decreased its autokinase activity, we hypothesized that Prc will modulate VgrR–DNA-binding affinity.

Chromatin immunoprecipitation sequencing (ChIP-seq) was employed to compare the VgrR-binding promoters in the genome of *X. campestris* pv. *campestris* 8004. As shown in Fig. 7a and Supplementary Table 4, after being challenged by 1.0 M sorbitol, ChIP-seq analysis revealed that the VgrR of the WT strain was potentially bound to the promoters of 87 genes (GenBank GEO:

GSE120292). In the Δprc mutant, VgrR bound the promoters of 105 genes. Only 41 VgrR-regulated genes were shared by the two strains under osmostress (Fig. 7a). *prc* may regulate VgrR-promoter binding, and, therefore, a *prc* deletion may cause changes in genome-scale binding events. The predicted VgrR-binding DNA motif is highly similar to that previously described (Fig. 7b)[25]. All of the identified VgrR-regulated genes were functionally classified into 16 categories (Fig. 7c, Supplementary Table 4). Among them, the osmostress-related genes, such as those involved in lipopolysaccharide synthesis (*XC_3814*), lipoic acid synthesis (*XC_0713*) and transportation (*XC_1619*, *XC_0969* and *XC_4223*), were absent in the Δprc mutant's dataset. Notably, The *prc* promoter was identified by ChIP-seq analysis (Supplementary Table 4). The promoter region of *prc* also contains a VgrR-binding motif that was then verified by an in vitro electrophoretic mobility shift assay (EMSA) (Fig. 7d). In the WT strain, the *prc* mRNA level was significantly induced after sorbitol stimulation (Fig. 7e), and in vivo ChIP, together with a quantitative PCR (ChIP-qPCR) assay, revealed that more VgrR was bound to the promoter regions of *prc* under stress conditions (Fig. 7f). However, in both *vgrR* and *prc* mutants containing 669–2,139-bp deletions of *prc*, *prc* (or remaining *prc* sequence, 497–665 bp) mRNA levels were significantly decreased, regardless of the osmostress stimulation (Fig. 7e). In the *prc* mutant, the level of VgrR occupancy on the *prc* promoter was significantly decreased, while genetic complementation significantly increased the level of VgrR occupancy (Fig. 7f). Thus, the transcription of *prc* was directly regulated by VgrR, which may form a feedback regulatory loop between *vgrS–vgrR* and *prc*.

We selected six genes from the ChIP-seq data, *XC_0690* (encoding a sugar kinase), *XC_0943* (conserved hypothetical protein), *XC_2164* (YciE orthologue of *E. coli* that is involved in osmotic stress responses)[28], *XC_3300* (outer-membrane protein) *XC_3301* (oxidoreductase), and *XC_3576* (outer-membrane protein), for further functional investigation. As shown in Fig. 8a, independent deletions of *XC_0943*, *XC_2164*, *XC_3300* and *XC_3301* resulted in decreases in bacterial growth on NYG plus 1.0 M sorbitol plates, while the inactivation of *XC_0690* and *XC_3576* did not have any recognizable impacts, suggesting that *XC_0943*, *XC_2164*, *XC_3300* and *XC_3301* are involved in the

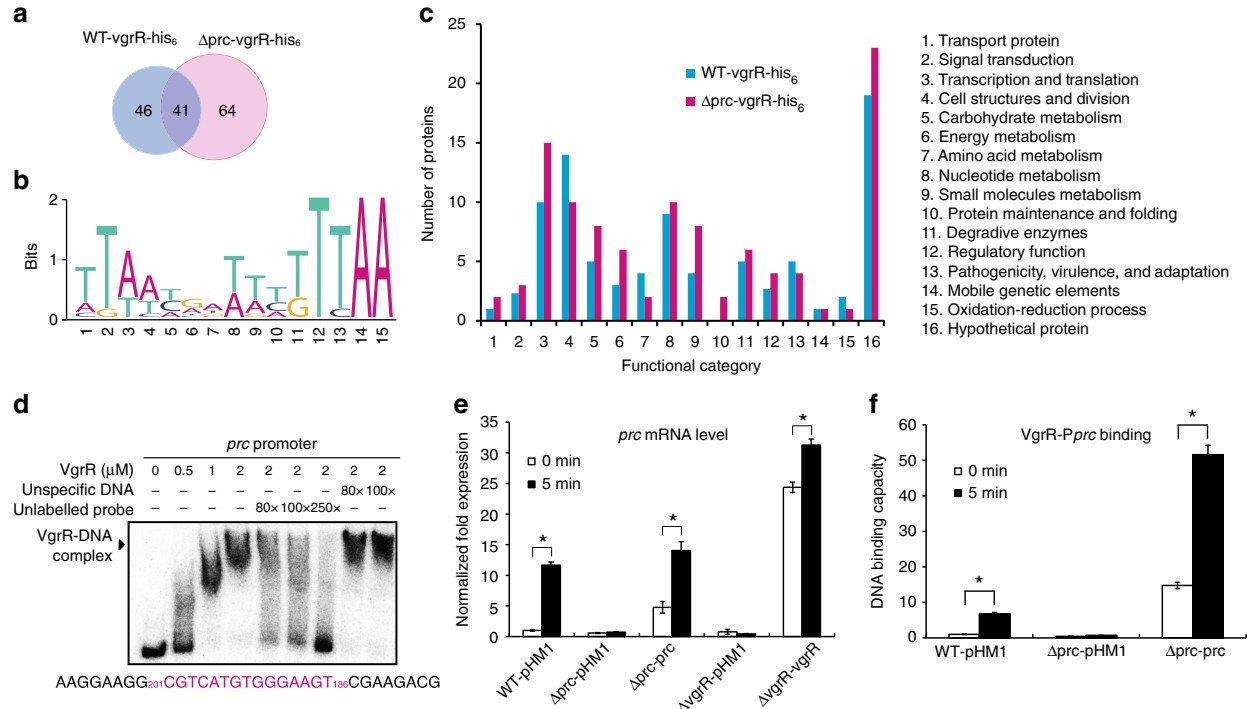

**Fig. 7** *prc* controls the VgrR regulon and VgrR promoter-binding landscapes. **a** Venn diagram showing the number of genes with promoters that potentially bound to VgrR. ChIP-seq was used to identify the VgrR-binding DNAs. Genes identified from the WT strain and the *prc* mutant are shown. **b** Predicted consensus VgrR-binding DNA motif based on ChIP-seq data. Weblogo was used to show the nucleotide composition. **c** Functional classification of the VgrR-regulated genes identified by ChIP-seq. Gene details are listed in Supplementary Table 4. **d** An electrophoretic mobility shift assay revealed that VgrR directly bound the promoter region of *prc*. PCR products of the promoter region were labelled with [γ-$^{32}$P]ATP and used as DNA probes. Unlabelled DNA and unspecific DNA were used as competitors. The sequence of the DNA probe is shown below, with the VgrR-binding motif in magenta. Numbers indicate the location relative to the translation initiation site. Each experiment was repeated two times. Triangle indicates VgrR–DNA complexes. **e** *vgrR* positively controls the transcription of *prc*. qRT-PCR was used to quantify *prc* mRNA in different bacterial strains before and after osmostress stimulation (1.0 M sorbitol, 5 min). Amplification of the cDNA of tmRNA was used as an internal control. A representative of three independent experiments is shown. **f** Deletion of *prc* decreases VgrR-promoter binding in bacterial cells. ChIP-qPCR was employed to quantify the enrichment of VgrR at the promoter region of *prc* under osmostress growth conditions (NYG medium plus 1.0 M sorbitol for 5 min) or no stimulation. The experiment was repeated three times. In **e** and **f**, error bars indicate the standard deviations. Asterisks indicate significant differences relative to the WT strain (Student's *t*-test, $P < 0.05$)

osmostress response. The EMSA revealed that VgrR physically bound to the promoter regions of these four genes (Fig. 8b, c, Supplementary Fig. 8). Competition upon the addition of various concentrations of unlabelled DNA probes gradually decreased the isotopic signals representing the VgrR–DNA complexes, demonstrating that VgrR directly binds the promoters of these genes. To investigate the role of Prc in controlling the transcription of these VgrR-regulated genes, a qRT-PCR analysis revealed that in the WT strain, sorbitol treatment significantly increased the expression levels of the four genes to 158% (*XC_0943*), 547% (*yciE*), 412% (*XC_3300*) and 174% (*XC_3301*) of the untreated sample (Fig. 8d, e, Supplementary Fig. 8). However, in the *prc* or *vgrR* mutants, stress-induced increases in the mRNA levels of these genes were not seen, whereas genetic complementation by overexpressing *prc* or *vgrR* fully restored deficiencies in the upregulation of the four genes (Fig. 8d, e, Supplementary Fig. 8). This result demonstrated that the inactivation of *prc* caused decreases in the mRNA levels of the four genes relative to the WT strain.

*Prc* control of the transcription levels of VgrR-regulated genes led to the hypothesis that in bacterial cells, proteolysis of VgrS by Prc decreased the phosphorylation level of VgrR, which modulated the TF activity of VgrR. MST analyses then revealed that unphosphorylated VgrR bound to the promoter regions of P*prc*, P*XC_0943*, P*XC_2164*, P*XC_3300* and P*XC_3301* with $K_d$ values from 0.46 to 7.62 μM (Supplementary Fig. 9). However,

when VgrR was phosphorylated by VgrS in the presence of ATP, VgrR-P did not bind to the promoter regions of P*prc*, P*XC_0943*, P*XC_2164* or P*XC_3301*, and the binding affinity between VgrR-P and P*XC_3300* was decreased ($K_d = 30.7$ μM). When inactive VgrS was added to the reaction, the binding affinities between VgrR and DNA were similar to those of unphosphorylated VgrR (Supplementary Fig. 9). Thus, unphosphorylated VgrR bound the promoter regions of downstream genes in vitro with higher affinity levels than those of phosphorylated VgrR.

To determine the changes in VgrR–DNA-binding affinities in vivo, ChIP-qPCR was used to estimate the levels of VgrR occupancy in the promoter regions of the selected genes, *XC_0943* and *yciE*. As shown in Fig. 8f, g, after stimulation by osmostress, the amounts of promoter DNA bound by VgrR were significantly increased in the WT strain to 1110% (P*XC_0943*) and 362% (P*yciE*) relative to the unstimulated samples. However, in the *prc* mutant, the amounts of promoter DNA bound to VgrR after stress stimulation were significantly decreased relative to the levels in the WT strain. In the genetic complementation strain of the *prc* mutation, in which *prc* was overexpressed, levels of VgrR–DNA binding were restored to, or even higher than, WT levels (Fig. 8f, g). Therefore, these genetic and biochemical analyses revealed that Prc positively regulates the binding between VgrR and the promoters of these osmostress response genes.

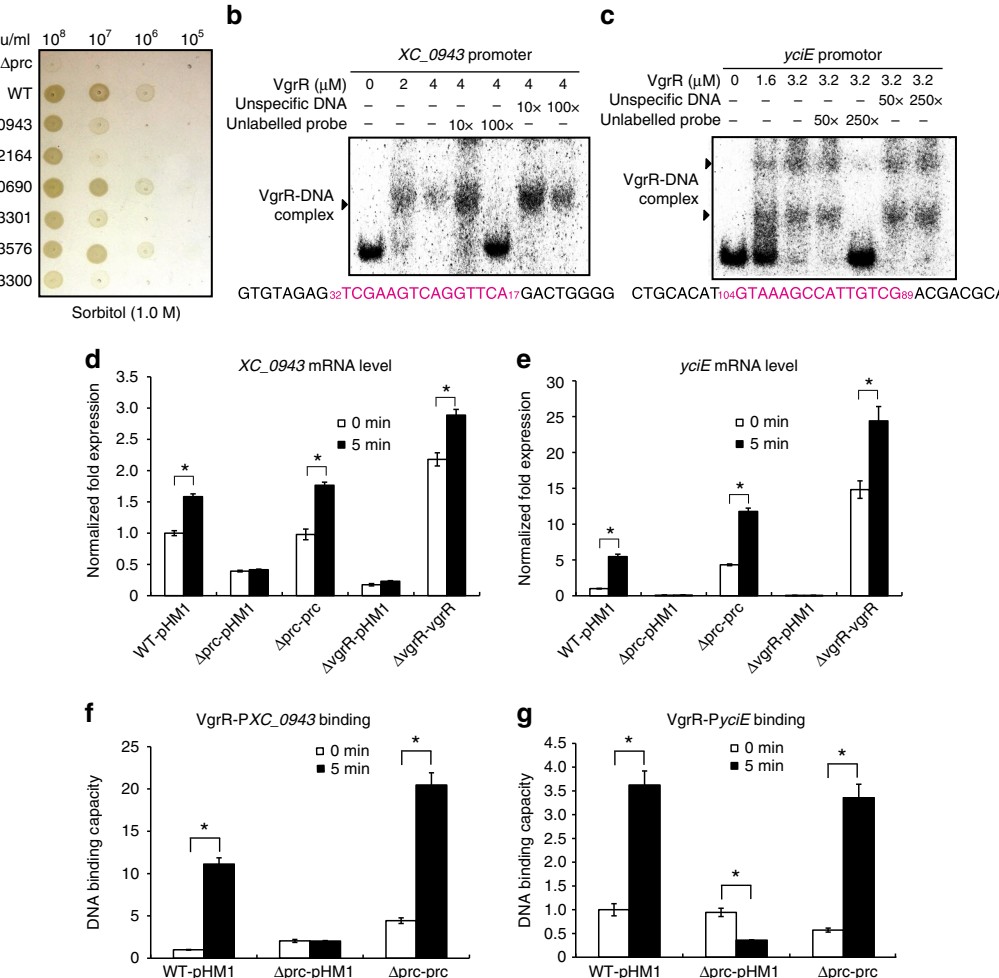

**Fig. 8** Prc regulates the VgrR–promoter-binding interactions in bacterial cells. **a** Bacterial growth on NYG–sorbitol plate. Bacterial cultures were serially diluted and inoculated onto NYG plus 1.0 M sorbitol plate and grown for 72 h at 28 °C. The experiment was repeated three times. **b** and **c** Electrophoretic mobility shift assays revealed that VgrR directly bound the promoter regions of *XC_0943* and *yciE*. PCR products of the promoter regions were labelled with [γ-$^{32}$P]ATP and used as DNA probes. Unlabelled DNA and non-specific DNA were used as competitors. The sequence of the DNA probe is shown below with the VgrR-binding motif in magenta. Numbers indicate the location relative to the translation initiation site. Each experiment was repeated two times. Triangles indicate VgrR–DNA complexes. **d** and **e** The *prc* mutation caused decreases in the transcription levels of *XC_0943* and *yciE* when bacterial strains were grown under osmostress. qRT-PCR was used to quantify the mRNA levels of these genes in different bacterial strains before and after osmostress stimulation (1.0 M sorbitol, 5 min). Amplification of the cDNA of tmRNA was used as an internal control. A representative of three independent experiments is shown. **f** and **g** The *prc* mutation caused decreases in VgrR–DNA binding in bacterial cells. ChIP-qPCR was conducted to quantify the enrichment of VgrR at the promoter regions of *XC_0943* and *yciE* in vivo when bacterial strains were grown under osmostress conditions (NYG medium plus 1.0 M sorbitol for 5 min). The experiment was repeated three times. In **d**–**g** Error bars indicate the standard deviations. Asterisks indicate significant differences of strains before and after osmostress (Student's *t*-test, *P* < 0.05)

## Discussion

Bacterial cells, except those of *Mycoplasma* spp., usually encode several to over a hundred HKs to detect various environmental stimuli[29,30]. Because HKs react to stimuli mainly through autophosphorylation and catalyse phosphoryl transfer to their cognate RRs, which control multiple cellular responses, regulation of the tempo and mode of HK phosphorylation is important for bacterial adaptation[31]. Chemical ligands, environmental cues, phosphatases (including cognate RRs), heterogeneous HKs and auxiliary proteins directly adjust the phosphorylation levels of HKs[32,33]. All of these regulatory modes are reversible in controlling HK phosphorylation. However, irreversible proteolytic modifications of HKs are less well-studied. In the present work, we demonstrated that under osmostress, a Prc protease of *X. campestris* pv. *campestris* specifically cleaved the periplasmic sensor region of an HK, VgrS, and inhibited its autokinase

activity. Quenching VgrS phosphorylation then modulated the TF activity of VgrR to bind the promoters of downstream genes. In addition, the transcription of *prc* itself was subject to regulation by VgrR, which then formed a positive feedback loop between the protease and the RR. This signalling process significantly and specifically promoted bacterial resistance to osmostress (Fig. 9). Thus, proteolysis of the sensor region of HK appears to be a molecular mechanism for regulating bacterial TCS phosphorylation and has an unambiguous physiological impact on bacterial adaptation.

Although studies investigating the relationship between proteases and HKs are lacking in bacteria, the proteolysis of Thr/Ser/Tyr receptor kinases to modulate their activities has been extensively studied in eukaryotic cells[5,34,35]. In Proteobacteria, Prc/Tsp-family proteases are widely distributed. These proteases not only recognize and process the C-termini of protein

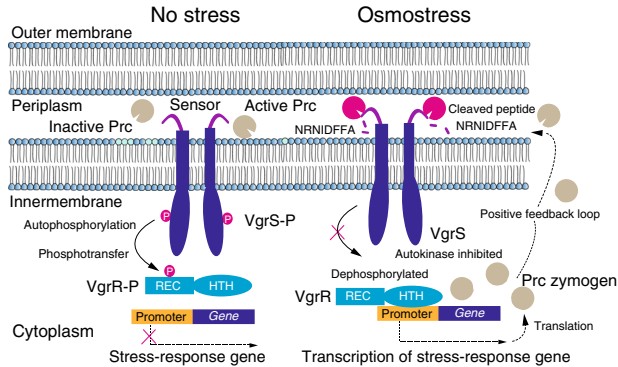

**Fig. 9** A model for VgrS proteolysis-triggered bacterial stress responses. When bacteria grow under osmostress, Prc is activated and then cleaves the N-terminal peptide (NRNIDFFA) of the sensor region of VgrS to inhibit the latter's autophosphorylation. Dephosphorylation of the VgrS–VgrR two-component signal transduction system regulates VgrR–promoter interactions and triggers the transcription of stress-response genes, which is required for *X. campestris* pv. *campestris* to resist osmostress. *prc* itself is controlled by VgrR, resulting in a positive feedback loop within the regulatory cascade

substrates but also degrade the substrates into small peptides[36,37]. For example, the Prc of *E. coli* controls cell wall synthesis and antibiotic resistance by degrading peptidoglycan hydrolases or outer-membrane lipoproteins[38,39]. The results of our study indicated that the cleavage of HK sensors by proteases localized in periplasmic space, including Prc/Tsp-family (MEROPS S41A-family) and HtrA-family proteases (MEROPS S1-family)[40], is important for regulating bacterial adaptation because, in gram-negative bacteria, a majority of the HKs contain periplasmic sensors to detect environmental signals[33]. Compared with reversible dephosphorylation, proteolysis of the HK sensor holds advantages in regulating TCS signalling: cleavage takes place in the periplasm, rather than the cytoplasm, which is more efficient because it shortens the distance travelled by TM signals. In particular, irreversible proteolysis avoids the continuing phosphorylation of existing HKs, which is more effective if reversible phosphorylation is extremely detrimental to bacterial survival in certain circumstances[1,17]. However, more experimental studies are needed for the following reasons: (1) there is no consensus recognition or processing sequence for the Prc/Tsp-family and HtrA-family proteases, which makes it difficult to predict the physiological substrates of these proteases. (2) as our results revealed, proteolytic processing of the VgrS sensor by the Prc protease is specific and adaptive to the osmostress response (Fig. 6), suggesting that protease–HK interactions are highly specific to environmental signals.

VgrS–VgrR is a pleiotropic TCS of *X. campestris* pv. *campestris*. It controls virulence, multiple-stress responses, bacterial growth and the capability of eliciting hypersensitivity reactions in non-host plants[25,26]. The VgrS sensor contains an EPQE motif to directly bind ferric iron. In iron-depleted environments, VgrS maintains a high level of autophosphorylation and then phosphorylates VgrR to regulate the expression of virulence factors[25]. Substitutions of the conserved phosphorylation sites in VgrS (His[186]) or VgrR (Asp[51]) caused decreases in virulence against host cabbage. However, here, in a *prc* mutant that exhibited a growth deficiency under osmostress, various mutations of *vgrS*, including deletion of the N-terminal sequence that mimics cleavage by Prc, effectively suppressed the growth deficiency caused by the null mutation of *prc* (Fig. 6h). In addition, substitution of the conserved His[186] of VgrS or the Asp[51] of VgrR (VgrR[D51A])

increased the bacterial growth rate under stress conditions (Fig. 6f). Thus, a low level of VgrS phosphorylation may promote bacterial resistance to osmostress, and the inhibition of VgrS autokinase activity by Prc proteolysis is an effective mechanism. In bacteria, a number of RRs are degraded by proteases in the cytosol, such as CtrA of *Caulobacter crescentus* and DegU of *Bacillus subtilis*[41,42]. Compared with the proteolysis of RRs, the cleavage of HKs by proteases may be more flexible in TCS regulation because of the following: (1) HK is a receptor, rather than an effector, in cell signalling[43]. Thus, degradation of the HK only impacts the phosphorylation level of the cognate RR (effector). This does not cause complete inactivation of the TCS because the dephosphorylated RR can continue to regulate physiological processes. (2) Bacteria usually encode paralogous TCSs that are generated by operon duplication during evolution[44]. These TCSs form regulatory networks, and the proteolysis of an HK reinforces the signalling of the existing compensatory pathways of these paralogous TCS. This is probably important for the VgrS–VgrR system because there are two functionally unknown, paralogous TCSs in *X. campestris* pv. *campestris* (XC_3126-XC_3125 and XC_3452-XC_3451)[25]. (3) The phosphatase activity of HK is essential for maintaining high levels of specificity between cognate HKs and RRs[45,46]. The proteolysis of an HK releases the inhibition on the illegitimate phosphorylation of the RR and then provides an opportunity for cross-talk among different TCSs. Future experiments are needed to verify these possibilities.

Our proteolytic experiment showed that Prc cleaved and degraded the purified VgrS sensor in a low-velocity in vitro experiment (~60 min, 28 °C, Fig. 4d), which seemed slower than the cleavage of full-length VgrS in vivo (~5–10 min, 28 °C, Fig. 4f). This difference indicated that the purified VgrS sensor may undergo a conformational change that interferes with Prc recognition in vitro. Alternatively, other adaptors or cellular factors are involved in Prc proteolysis within bacterial cells. Adaptors are proteins that bind proteases to help with the recognition of substrates or proteolysis. In *E. coli*, NlpI is an adaptor of Prc that helps facilitate the protease's degradation of its physiological substrate, MepS, an outer-membrane protein that is associated with peptidoglycan biosynthesis[47]. The presence of NlpI enhances Prc protease activity by 50-fold[48]. Our TAP analysis provided information on Prc-binding proteins, but it is difficult to predict which one is the Prc adaptor used during processing of the VgrS sensor because adaptors are diverse in structure and include lipoproteins, RRs and arginine kinases[49]. In addition, how the Prc protease is activated to cleave the VgrS sensor remains unclear. Our preliminary experiment showed that a high concentration of sorbitol (200 mM to 1.0 M) did not affect the enzymatic activity of Prc, indicating that Prc itself is not a sensor that directly detects osmostress. As Fig. 3c revealed[48], Prc exists as a monomer or trimer in vitro, and only monomeric Prc bound and degraded the VgrS sensor (Fig. 3c, e). This implied that alteration of the quaternary structure of Prc might act as a molecular mechanism to activate its protease activity in vivo.

In addition, the regulatory functions of the proteolytic products of VgrS are unknown. In eukaryotes, fragments of receptor kinases generated by proteases are usually stably maintained in cells[50,51]. These fragments serve as dominant negative factors, decoy receptors or even cellular signals[52,53]. As Fig. 5 and Supplementary Fig. 6 show, Prc proteolysis produced a short N-terminal peptide and a truncated VgrS. The two fragments were detected by western blotting and MS analysis, and they were relatively stable. In TCS regulation, the majority of HKs constitutively form homodimers that are autophosphorylated in a *trans* manner[33]. The presence of a truncated, "decoy" VgrS may disturb this process. Furthermore, if VgrS and its paralogous HKs (XC_3125 and XC_3451) constitute regulatory heterodimers,

VgrS proteolysis could affect the signalling of other TCSs. The biological roles and fates of the two fragments remain unclear. Future investigations are needed to study the activating mechanism of Prc under osmostress and the regulatory roles of HK fragments. The results will provide insights into the cross-talk between TCS signalling and regulation through proteolysis.

## Methods

**Bacterial strains and growth conditions.** Strains used in this study are listed in Supplementary Table 1. *E. coli* strain BL21 (DE3) (Novagen) was used to express recombinant proteins, and *E. coli* strain DH5α (Lab collection) was used for molecular cloning. The *E. coli* strain was cultured at 37 °C in Luria–Bertani medium supplemented with the appropriate antibiotics. *Xanthomonas campestris* pv. *campestris* 8004 was cultured at 28 °C in NYG medium with the appropriate antibiotics. *Xanthomonas oryzea* pv. *oryzea* PXO99 was cultured at 28 °C in PS medium with the appropriate antibiotics. The following concentrations of antibiotics were used: 100 μg/ml ampicillin, 50 μg/ml kanamycin, 100 μg/ml spectinomycin, 34 μg/ml chloromycetin and 25 μg/ml rifampicin. The transformation of bacterial competent cells was performed according to previous studies[18,25]. Bacterial electro-competent cells were prepared by extensively washing fresh bacterial cells for three times with ice-cold glycerol (10%). Transformation condition of bacterial competent cells were set as 1.6 kV cm$^{-1}$ to 2.0 kV cm$^{-1}$ and conducted in a Bio-Rad Pulser XCell (Bio-Rad, USA).

**Construction of bacterial recombinant strains.** In-frame deletion mutants and recombinant strains of *X. campestris* pv. *campestris* were constructed using the homologous, double-cross-over method with the suicide vector pK18mobsacB as described in a previous study[25]. Briefly, the upstream and downstream genomic sequences of the region to be deleted were amplified using primers listed in Supplementary Table 2. After restriction enzyme digesting and ligating into pK18mobsacB, the recombinant plasmid was electroporated into competent cells to generate single-cross-over mutants by selection on NYG plates containing kanamycin. The putative single-cross-over mutants were confirmed by PCR, cultured in NYG medium for second-round homologous cross-over, and grown on NYG plates containing 10% sucrose to select in-frame deletion mutants, which were further verified by PCR and subsequent sequencing. Genetic complementation was achieved in *trans* by providing full-length genes using the broad host vector pHM1.

**Phenotypic characterization.** Plant inoculations and virulence assays were performed as described previously[26]. Briefly, strains of *X. campestris* pv. *campestris* were inoculated into susceptible 4-week-old cabbage (*Brassica oleraceae* cv. Jing-Feng 1) hosts by clipping the leaves with pre-sterilized scissors dipped into the bacterial culture. The lesion level was measured at 10 d after inoculation with at least 12 repetitions. Virulence level was scored by a semi-quantitative standard as previously reported[19,54]: 0-level, no visible effect; 1-level, limited chlorosis around the cut site; 2-level, chlorosis extending from the cut site; 3-level, blackened leaf veins, death and drying of tissue within the chlorotic area; 4-level, extensive vein blackening, death and drying of tissue. For *in planta* growth assay, bacterial strains were infiltrated into plant leaves by a 1-ml syringe and leave disks (diameter = 0.8 cm) were periodically collected, ground in 1 ml NYG medium, and serially diluted before plating and colony counting. For each assay, bacterial population in six leave disks was determined. Bacterial stress resistance assays were performed on NYG plates supplemented with 2.5 mM FeSO$_4$ (plus 5 mM vitamin C) and 1.0 M sorbitol. Strains were cultured to 10$^6$ CFU/ml (OD$_{600}$ = 0.4) in NYG medium with the appropriate antibiotics, and then 1 μl of 10-fold serial-dilution cultures were inoculated onto the plates and grown for 2–3 d at 28 °C.

To determine the bacterial sensitivity to antibiotics (erythromycin and kanamycin), strains were cultured to 10$^6$ CFU/ml (OD$_{600}$ = 0.4) in NYG medium. Then, 1 ml of culture was mixed with 30 ml of melted NYG agar plate, just before the agar solidified. After the plate was fully solidified, filter paper disks (Φ = 0.6 mm) containing 5 μl erythromycin (20 μg/ml) or kanamycin (50 μg/ml) were placed on the plate to inhibit bacterial growth. Minimal inhibition concentration determinations were performed using the standard serial-dilution method[55].

**Protein expression and purification.** The pET System (Novagen, Madison, WI, USA) was used for the cloning and expression of recombinant proteins in *E. coli*. Recombinant vectors used to express proteins are listed in Supplementary Table 1. Induction, expression, and purification were performed using affinity chromatography with Ni-NTA agarose beads (Novagen, USA) according to the manufacturer's manual. Purified His$_6$-tagged proteins were concentrated with Amicon Ultra-4 and -15 Centrifugal Filter Units (Merck Millipore, Germany) and dissolved in storage buffer (50 mM Tris-HCl, pH 8.0, 0.5 mM EDTA, 50 mM NaCl, and 5% glycerol) before use.

**Gel filtration chromatography.** The purified proteins were concentrated with Amicon Ultra-4 and -15 Centrifugal Filter Units and subsequently subjected to gel filtration with Fast Protein Liquid Chromatography AKTA Purifier 10 with Frac-

900 (GE Healthcare, USA). The ATKA system was pre-equilibrated with 20 mM Tris-HCl, pH 8.0, at a flow rate of 0.5 ml/min and then applied to a Superdex 75 10/300 GL column to separate dimers and oligomers from the monomer. The elution profiles were collected at A$_{280}$ and confirmed by SDS-PAGE gels and western blots.

**Analytical ultracentrifugation.** Analytical ultracentrifugation was performed at 20 °C in a ProteomeLab XL-1 analytical ultracentrifuge (Beckman, USA) equipped with absorbance optics and an An60-Ti rotor. The sedimentation velocity analysis of the affinity chromatography-purified Prc$^{S475A}$ was conducted at 240,000 g using double sector cells, and data were collected at the concentration of A$_{280}$ = 0.7 in sodium phosphate buffer (pH 8.0) plus 150 mM NaCl for each peak sample. These values were normalized to standard conditions by correcting for buffer density and viscosity. Interference sedimentation coefficient distributions were calculated from the sedimentation velocity data using the SEDFIT software program (www.analyticalultracentrifugation.com).

**Subcellular cell fraction.** To prepare the total cell protein fraction, the bacterial cells were collected and resuspended in 1X SDS-PAGE loading buffer (50 mM Tris-HCl, pH 6.8, 2% SDS, 0.1% bromophenol blue, 10% glycerol and 100 mM DTT) and immediately boiled for 3 min to denature the protein.

To prepare the periplasmic fraction, the bacterial cell pellets were thoroughly resuspended in Buffer A (30 mM Tris-HCl, pH 8.0, 20% sucrose and 1 mM EDTA). After rotating at room temperature for 10 min, the cells were centrifuged and the cell pellet was resuspended in ice-cold Buffer B (5 mM MgSO$_4$). After rotating at 4 °C slowly for 10 min on ice, the periplasmic proteins were released into the buffer. Finally, the periplasmic fraction was concentrated by ultrafiltration. To collect the soluble cytoplasmic fraction, the bacterial cells were collected and resuspended in protein lysis buffer [50 mM Tris-HCl, pH 7.5, 0.5 M NaCl, 10% glycerol, 0.1% Triton X-100 and 1 mM phenylmethylsulfonylfluoride (PMSF)]. After sonicating, the supernatant represented the cytoplasmic fraction.

To isolate the inverted membrane vesicles, the bacterial cells were collected, resuspended and sonicated in a low-salt buffer (100 mM sodium phosphate, pH 7.0, 10% glycerol, 5 mM EDTA, 10 mM DTT, 1 mM PMSF). The inverted membrane vesicles were collected by ultracentrifugation (150,000 × *g* for 60 min at 4 °C) before removing the unbroken cells and debris through two rounds of centrifugation (8000 × *g* for 10 min at 4 °C). After washing with high salt buffer (20 mM sodium phosphate, pH 7.0, 2 M KCl, 10% glycerol, 5 mM EDTA, 5 mM DTT and 1 mM PMSF), the inverted membrane vesicles were resuspended in storage buffer (20 mM Tris-HCl, pH 7.5 and 10% glycerol).

**Enzymatic and endo-protease activity assays.** The qualitative enzymatic activity assay was performed in the enzymatic reaction buffer (50 mM Tris-HCl, pH 7.5, 2 mM DTT, 25 mM NaCl, 25 mM KCl and 5 mM MgCl$_2$) with the proper amounts of protease and substrate co-incubated at 28 °C for the indicated time, or at a different temperature, DTT concentration, or pH value for 5 min. The reactions were stopped by adding SDS-PAGE loading buffer before loading the samples into a 12% PAGE gel and performing the electrophoresis. The gel was stained with Coomassie brilliant blue to determine the remaining substrate.

The quantitative *endo*-protease activity assay was carried out using azocasein (Megazyme) as the substrate. The pre-equilibrated protease (5 μM) was co-incubated with pre-equilibrated azocasein (400 μM) in Buffer A (100 mM sodium phosphate). The solution was fully mixed and incubated for 10 min at 37 °C. The enzymatic reaction was terminated by the addition of 2.5 times the volume of 5% trichloroacetic acid to precipitate the non-hydrolyzed azocasein. The reaction was vigorously stirred for 5 s and stood for 5 min at room temperature. Then, it was centrifuged at 3000 × *g* for 10 min, and the absorbance of the supernatant solutions at 440 nm was determined. The samples, minus the protease, which was substituted with Buffer A, served as blanks.

To measure the enzyme kinetics of the protease, the same amount of protease was co-incubated with different amounts of azocasein in Buffer A (100 mM sodium phosphate). The subsequent procedure was performed in the same manner as the qualitative *endo*-protease activity assay. The $K_M$, which is the Michaelis–Menten constant, and $V_{max}$, which is the maximum reaction velocity, values were determined using the Lineweaver–Burk plot method.

**TAP and MS identification.** TAP was carried out following the protocol of the FLAG HA Tandem Affinity Purification Kit (Sigma-Aldrich). Briefly, the bacterial cells were collected and ground before being resuspended in lysis buffer [50 mM Tris-HCl, pH 8.0, 0.15 M NaCl, 1 mM EDTA, pH 8.0, 1 mM PMSF and 1 tablet/20 ml protease inhibitor cocktail tablets (Roche)]. They were centrifuged at 13,000 × *g* for 20 min at 4 °C, and the supernatant represented the whole cell lysate. Pre-washed ANTI-FLAG M2 resin was added to the sample lysate and incubated from 2 h to overnight at 4 °C with slow shaking. For each wash, the resin was gently agitated in the lysis buffer, centrifuged at 3000 × *g* for 1 min, and then the remaining final wash volume was decanted without losing any resin. The resin–protein complex was washed with lysis buffer using three rounds of low-speed centrifugation. The remaining resin was transferred into a spin column, and 2.5 volume of 3X FLAG peptide (150 ng/μl) was added and co-incubated for at least

10 min at 4 °C. The column was spun before removing the tip to a clean micro-centrifuge tube and keeping the eluate, which contained the eluted FLAG–protein complex. The elution process was repeated twice. Then, 40 μl pre-washed ANTI-HA resin in lysis buffer was added and incubated 30 min to 2 h at 4 °C with slowly shaking. The supernatant was removed after the incubation, and the ANTI-HA resin–protein complex was washed with lysis buffer through three rounds of low-speed centrifugation. Then, 50 μl 8 M urea was added and co-incubated for a minimum of 10 min at room temperature. The sample was centrifuged at $3000 \times g$ for 1 min, and the eluate was loaded and separated on a 12% SDS-PAGE gel. After silver staining, the differential gel bands between sample and control were manually excised, and each band then underwent enzymatic digestion.

The protein bands in the gel were further cut into small plugs, washed in 100 μl freshly made 30 mM $K_3Fe(CN)_6$ and 100 mM $Na_2S_2O_3$ (1:1, v-v) buffer for seconds, rinsed with 25 mM $NH_4HCO_3$, dehydrated in 100% acetonitrile for 10 min, and dried in a SpeedVac (Labconco) for 15 min. The reduction procedure was performed in solution A (10 mM DTT and 25 mM $NH_4HCO_3$) for 45 min at 56 °C, and then, the alkylation procedure was performed in solution B (40 mM iodoacetamide and 25 mM $NH_4HCO_3$) for 45 min at room temperature in the dark. The gel plugs were washed with 50% acetonitrile in 25 mM ammonium bicarbonate twice. The gel plugs were then dried in the SpeedVac for 15 min and digested with sequence-grade modified trypsin (50 ng per band) in 25 mM $NH_4HCO_3$ overnight at 37 °C. Formic acid was added to a 1% final concentration to stop the enzymatic reaction before transferring the solution to a sample vial for subsequent LC–MS/MS analysis.

The nanoLC–MS/MS identification of proteins was performed on a Thermo Finnigan LTQ linear ion trap mass spectrometer in line with a Thermo Finnigan Surveyor MS Pump Plus HPLC system. Tryptic peptides were loaded onto a trap column (300 SB-C18, 5 × 0.3 mm, 5-μm particle; Agilent Technologies, Santa Clara, CA, USA). The peptides were eluted over gradient solution C (80% acetonitrile and 0.1% formic acid) at a flow rate of 500 nl/min and introduced into the online linear ion trap mass spectrometer (Thermo Fisher Corporation, San Jose, CA, USA) using nano electrospray ionization. The five most abundant ions (one microscan per spectra) were selected for fragmentation from a full-scan mass spectrum by collision-induced dissociation for data-dependent scanning.

MS data were analysed with SEQUEST against NCBI's *X. campestris* pv. *campestris* 8004 protein database and displayed with Bioworks 3.2. Peptides with +1, +2, or +3 charge states and with cross correlations of ≥1.90, >2.5, and >3.0, respectively, were accepted. Carbamidomethylation on cysteine and oxidation on methionine were selected as residue modifications. SEQUEST was searched with a peptide tolerance of 3.0 Amu and a fragment of 1.0 Amu.

**Microscale thermophoresis (MST) measurement**. The MST measurement was used for detecting protein–protein interactions. Briefly, 10 μM purified VgrS sensor was labelled with a Monolith NT Protein Labeling Kit RED-NHS (Nano Temper Technologies GMBH, München, Germany) using red fluorescent dye NT-647 *N*-hydroxysuccinimide (amine-reactive) according to the manufacturer's instructions. The excess labelling reagents were removed by buffer-exchange column chromatography, the labelled VgrS sensor was eluted in NTA buffer (300 mM NaCl and 50 mM sodium phosphate buffer, pH 7.0). The binding assays were performed on a Monolith NT.115 Microscale Thermophoresis instrument (Nano Temper Technologies GMBH) using standardly treated capillaries. Equal amounts of labelled protein were titrated by the purified Prc$^{S475A}$-monomer, PrcΔPDZ$^{S475A}$, and PEP + DUFΔ20$^{S475A}$, which were exchanged into the 1X NTA buffer with 0.05% Tween, using a 1:1-series dilution method. The curves were fitted by Nano Temper Analysis Version 1.5.41 from three replicates, and the value of dissociation constant ($K_d$) was calculated.

MST was also used to quantify the VgrR–DNA binding affinities. 5′-FAM-labelled oligonucleotide primers for downstream genes were synthesized and used in PCR. The labelled double-stranded DNA product was added to serially diluted protein reaction volumes (at an initial concentration of 40 μM) containing 50 mM Tris-HCl (pH 7.4), 150 mM NaCl, 10 mM $MgCl_2$ and 0.05% (v/v) Tween-20. The KD Fit function of NanoTemper Analysis software version 1.5.41 was used for curve fitting and calculation of the value of the dissociation constant ($K_d$).

**SPR**. The binding kinetics of protein–protein interactions were estimated by the SPR assay, which was performed using a Biacore 3000 (GE Healthcare) at 25 °C. The running buffer was PBST (0.27 g/L $KH_2PO_4$, 1.42 g/L $Na_2HPO_4$, 8 g/L NaCl, 0.2 g/L KCl, pH 7.4 and 0.005% Tween-20). As the manufacturer described, one flow cell of a CM5 sensor chip was activated with a mixture of 0.2 M *N*-(3-dimethylaminopropyl)-*N*′-ethylcarbodiimide hydrochloride and 0.05 M *N*-hydroxysuccinimide in ddH2O. The ligand protein, in 10 mM sodium acetate (pH 4.5), was injected over the flow cell for 10 min at a flow rate of 10 μl/min. The remaining binding sites were blocked by 1 M ethanolamine (pH 8.5). In total, 3000 response units of the ligand protein were immobilized. Protein at different concentrations (double dilution) was injected into the immobilized protein and blank flow cells for 5 min at a flow rate of 30 μl/min. The bound protein was removed with 0.01% SDS for 30 s after the 5-min dissociation phase. The curves were fitted to the 1:1 Langmuir binding model or steady state affinity model (BIA evaluation 4.1 software) to obtain the equilibrium and kinetic constants.

**RT-PCR, qRT-PCR and western blotting**. The total RNA was extracted using TRIzol reagent (Invitrogen, Carlsbad, CA, USA), according to the manufacturer's instructions. RNA concentrations were determined using a NanoDrop 1000 spectrophotometer (Thermo Scientific, Wilmington, DE, USA). Then, 1 μg total bacterial RNA was treated with RNase-free DNAse (Ambion, Austin, TX, USA.) to remove contaminating DNA, and the first strand cDNA was synthesized with random primers (Promega, Madison, WI, USA) using Superscript III reverse transcriptase (Invitrogen).

To analyse the Prc operon's structure, with cDNAs as templates, primers were used to amplify the possible transcripts from the intergenic regions between genes. Real-time quantitative reverse transcription PCR (qRT-PCR) was used to quantify the mRNA levels. qRT-PCR was performed using Maxima SYBR Green (Fermentas, Glen Burnie, MD, USA) with a CFX96 real-time PCR detection system (Bio-Rad, Hercules, CA, USA). The transcript level of 16 s rRNA was amplified as the internal control. The qRT-PCR conditions were as follows: 95 °C for 10 min; 44 cycles of 95 °C for 15 s, 60 °C for 30 s, and 7 °C for 30 s; and 95 °C for 15 s. The melt curve was performed at 65–95 °C, in increments of 0.5 °C for 5 s.

Western blotting was conducted by transferring the proteins onto PVDF membrane (Millipore, USA). Polyclonal antiserum of VgrS or monoclonal antibodies of HA (M20003-L) and His$_6$ tags (M20001L, Abmart, China) were used to detect the proteins. All of the antibodies were diluted 5000–10,000-folds before use.

**Identification and quantification of proteolytic peptide by QTRAP LC–MS/MS**. For detecting the N-terminal proteolytic peptide of VgrS, Prc-treated VgrS inverted membrane vesicles were collected at 0, 5, 10, 30, 60, 120 and 180 min and immediately subjected to liquid nitrogen for fixing the proteolytic status. After centrifuging the mixer with 3K Ultrafiltration devices (Pall Corporation, USA) at $11,000 \times g$ for 10 min, the peptides below 3 kDa were concentrated for further analyses. The chemically synthesized VgrS N-terminal peptide (2nd to 9th aa) was used as a standard. The samples were analyzed on an Agilent 1200 pump LC system (Agilent Technologies, Santa Clara, CA, USA) equipped with a Synergy 4 μM Hydro-RP 80 Å LC column (150 × 2 mm, Phenomenex), which is coupled to a QTRAP 4500 Mass Spectrometer instrument with a TurboIonSpray ionization source (Applied Biosystems, AB SCIEX, CA, USA). For all the separations, the flow rate of the mobile phase was 400 μL/min., and eluent A and B of the mobile phase was water and 100% methanol, respectively. The injection volume was 3 μL. The gradient separation and elution was as follow: time = 0 min: 98% A, 2% B; 3 min: 98% A, 2% B; 24 min: 0% A, 100% B; 28 min: 0% A, 100% B; 29 min: 98% A, 2% B and 32 min: 98% A, 2% B. The ESI source was operated in positive ionization mode with dwell time of 150 ms for MRM analysis (461/120). The MS acquisition with the ESI interface was performed under the following condition: ion spray voltage, 4500 V; curtain gas 20 (arbitrary units); GS1 (Nebulizer Gas) and GS2 (Heater Gas), 60 psi, respectively; probe temperature, 500 °C. The selected reaction monitoring (SRM) measurements were performed using declustering potential (DP) of 80 V, entrance potential (EP) of 10 V, collision cell exit potential (CXP) of 13 V, and collision energy (CE) values of 18 optimized by the software. Evaluation of the data was performed with Analyst software (v.1.6.1) and the relative and quantitative amount of each sample was determined with Analyte Peak Area (counts).

**In vitro phosphorylation assay**. In the in vitro autokinase activity assay, the inverted membrane vesicles containing full-length wide-type or mutated VgrS were incubated with or without Prc or Prc$^{S475A}$ for 10 min at 28 °C, then incubated with 1 μl ATP containing 10 μCi [γ-$^{32}$P]-ATP (PerkinElmer) in 20 μl autophosphorylation buffer (50 mM Tris-HCl, pH 7.8, 2 mM DTT, 25 mM NaCl, 25 mM KCl and 5 mM $MgCl_2$) for 2 min. The phosphorylation reaction was stopped using 5X SDS-PAGE loading buffer. Ten micrometers of VgrR was added for phosphotransfer analysis if needed. The phosphorylated proteins were separated by 12% SDS-PAGE and exposed to a phosphor screen for 2–3 h. After being scanned using a PhosphorImage system (GE Healthcare, Chicago, IL USA), the gels were stained with Coomassie brilliant blue to determine the location and amount of the autokinase protein.

**EMSA**. PCR products of the promoter region were labelled with [γ-$^{32}$P]ATP using T4 polynucleotide kinase (Fermentas) and purified with a ProbeQuant G-50 column (GE, New York, NY, USA) that removed free [γ-$^{32}$P]ATP. The unlabeled DNA probe (0–10 μmol) and purified protein (0–10 μg) were incubated together for 30 min at 28 °C in reaction buffer (10 mM Tris-HCl, pH 7.5, 50 mM KCl, 1 mM DTT, 50 ng poly(dI:dC) and 10 mM EDTA, pH 8.0). Then, 10 fmol labelled-DNA probes were subsequently added to each reaction for binding and competition for another 30 min at 28 °C. To stop the EMSA reaction, 4 μl DNA loading buffer (0.25% bromophenol, 40% sucrose) was added. The samples were loaded onto a 5% native PAGE gel and performing the electrophoresis under 120 V for 40 min with 0.5X TBE buffer (5.4 g/l Tris, 2.75 g/l boric acid and 2 ml/l 0.5 M EDTA, pH 8.0) before autoradiography.

**ChIP-seq and ChIP-qPCR**. ChIP-seq was performed as previously reported[25,56]. Briefly, the bacterial strains grown in NYG medium until $OD_{600} = 0.8 \pm 0.05$ were treated with or without 800 mM sorbitol for 5 min. After collection by

centrifugation, the samples were crossing-linked with 1% (V/V) formaldehyde for 20 min and subsequently quenched with 20% (V/V) 0.5 M glycine for 10 min. Then, the bacterial cells were recollected, washed with cold PBS (0.27 g/l $KH_2PO_4$, 1.42 g/l $Na_2HPO_4$, 8 g/l NaCl, and 0.2 g/l KCl, pH 7.4) twice, and resuspended in ChIP lysis buffer (10 mM Tris, pH 8.0, 20% sucrose, 50 mM NaCl, 10 mM EDTA, pH 8.0, 10 mg/ml lysozyme, and 1 mM PMSF). Quadruple the lysis buffer volume of immunoprecipitation buffer (50 mM HEPES-KOH, pH 7.5, 150 mM NaCl, 1 mM EDTA, pH 8.0, 1% Triton X-100, 0.1% sodium deoxycholate and 0.1% SDS) was added to the bacterial cell suspension, and the suspension was sonicated with a Diagenode Bioruptor (Diagenode, Liège, Belgium), generating 100–300-bp DNA fragments. The cell lysis was pre-cleared with 20 µl protein A sepharose (GE) for 10 min at 4 °C on a rotator, and 300-µl aliquots were retained as the inputs. For the ChIP assay, 50 µl protein A sepharose and 2 µl anti-His$_6$ monoclonal antibody (Abmart, China, M20001L) was added to a 1.5-ml aliquot of the cell lysis. The mixture was incubated at 4 °C overnight or at least 8 h on a rotator. The protein A sepharose beads were collected and washed with immunoprecipitation buffer twice, and subsequently with wash buffer (10 mM Tris-HCl, pH 8.0, 250 mM LiCl, 1 mM EDTA, pH 8.0, 0.5% Nonidet-P40 and 0.5% sodium deoxycholate), high salt wash buffer (50 mM HEPES, pH 7.9, 500 mM NaCl, 1 mM EDTA, pH 8.0, 0.1% SDS, 1% Triton X-100 and 0.1% deoxycholate) and TE buffer (10 mM Tris-HCl, pH 8.0 and 1 mM EDTA, pH 8.0) once. After adding 100 µl of elution buffer (50 mM Tris-HCl, pH 7.5, 10 mM EDTA, pH 8.0 and 1% SDS) at 65 °C for 10 min, the immuno-precipitated chromatin DNA was removed from the beads. Then, the eluted DNA was treated with RNase A to remove contaminating RNA at 42 °C for 90 min and subsequently with proteinase K to remove contaminating protein at 65 °C over-night. Further, the DNA was extracted with 24:1 (v:v) chloroform:isoamyl alcohol, and precipitated with ethyl alcohol. Finally, DNA fragments of 100–500 bp were collected and purified using a 2% agarose gel and PCR purification kit (Qiagen, Duesseldorf, Germany), respectively. An Illumina HiSeq-2000 system (Illumina, San Diego, CA USA) was used for high-throughput sequencing, which was con-ducted by the Beijing Institute of Genomics genomic service. The high-throughput sequencing reads were analysed using the Burrows–Wheeler Aligner method, and the cleaned reads were aligned to the genomic sequence database of *X. campestris* pv. *campestris* 8004. Peak calling was conducted by MACS2[57]. The consensus binding motif analysis was completed with MEME and FIMI tools in the MEME software suite.

ChIP-qPCR was performed to quantify the amount of promoter DNA bound by the His$_6$-fused protein or VgrR in vivo. Input DNA (10 ng) was quantified with qPCR using Maxima SYBR Green (Fermentas). The percentage of immunoprecipitated promoter DNA was calculated in comparison to the amount of input.

**Reporting Summary**. Further information on research design is available in the Nature Research Reporting Summary linked to this article.

## Data availability

All relevant data in this work has been included in the manuscript. The original, uncropped images were shown in Supplementary Fig. 10. Data and experimental materials are available from the corresponding author upon request. The accession number for the ChIP-seq dataset reported in this study is GenBank GEO: GSE120292.

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

## Acknowledgements

The authors thank Profs. Yong Tao, George F. Gao and Ting-Yi Wen of our institute for valuable suggestions regarding the work. Dr. Xiang Ding and Dr. Xiao-Xia Yu of the Institute of Biophysics, Chinese Academy of Sciences helped to perform the MS and ultracentrifugation analyses. This work was supported by the Strategic Priority Research Program of the Chinese Academy of Sciences (Grant No. XDB11040700), the National Natural Science Foundation of China (Grants Nos. 31671989, 31600062 and 31370127), the Ministry of Science and Technology of China (Grant No. 2016YFD0100602) and the State Key Laboratory of Plant Genomics.

## Author contributions

W.Q. conceived the study. C.-Y.D. and W.Q. designed the experiments. C.-Y.D., H.Z., L.-L.D., Y.J.L., Y.P., S.-T.S., Y.W., L.W. and W.Q. performed the experiments and analysed the data. C.-Y.D. and W.Q. drafted the manuscript, and all authors contributed to the revisions. All authors read and approved the manuscript.

## Additional information

**Competing interests:** The authors declare no competing interests.

