## [Peer Review File · Nature Communications]

Reviewers' comments:

Reviewer #1 (Remarks to the Author):

The manuscript of Deng and co-workers reports a novel mechanism by which environmental signals control the activity of a two-component system. The authors argue that when the bacterium *X. campestris* experiences osmotic stress, the sensor protein VgrS is proteolysed by the protease Prc, which prevents VgrS autophosphorylation, therefore decreasing phosphorylation of VgrS cognate regulator VgrR, and abolishing transcription of VgrR-dependent genes. By contrast, in the absence of stress, Prc would not degrade VgrS resulting in VgrR phosphorylation and transcription of its target genes. Whereas proteases have been implicated in the stability of regulators belonging to the two-component system family, the proposed mechanism appears unique as the signal would, by virtue of proteolysing VgrS, render VgrS unable to conduct new rounds of autophosphorylation and phosphotransfer.

The manuscript provides a lot of information. However, a lot of it is unclear or missing controls. In addition, the writing is not clear. There are too many spelling and grammatical errors to list them individually. In addition, the authors use words to mean things other than their accepted meanings.

Below please find my specific comments.

Line 1: What do the authors mean by "sequesters"?

Line 29: What do you mean by "effective mechanism"?

Line 48: The sentence starting with "Cellular regulation..." is not clear.

Line 104: There is a discrepancy between the figure legend and the methods section. Did the authors use Fe₂SO₄ or Fe₂Cl?

Line 191: It is hard to follow the mutations on VgrS without a schematic of its predicted structure/conformation. The authors should refer to the location or amino acid region (as opposed to "sensor" domain).

Line 236: What is the autophosphorylation behavior of the A9Q Q10A VgrS double mutant?

Line 240: Contrary to what the authors state, amino acids 2 to 9 are not essential for autokinase activity, otherwise there would be no VgrS phosphorylation.

Line 249: The authors should provide growth data on media where there is no stress.

Line 278: The authors did not show dephosphorylation of VgrR.

Line 297: There are other interpretations for the findings.

Line 321: Are the authors really measuring the effects of three genes? They are examining the effect of prc inactivation and one of the genes being investigated is prc itself. Please explain.

Line 339: Please provide original citation for example of a given species encoding "hundreds of HKs".

Figure 4c: Degradation appears very inefficient.

Figure 5b: Differences are small.

Figure 5c: The *vgrS* and *vgrR* mutants behave differently from each other. Therefore, it is not clear what is really going on in terms of the relationship between phenotype and genotype. Moreover, the *vgrR* mutant specifying the D51A variant, which is predicted to not be subject to phosphorylation complements a Δ *vgrR* strain just like the wild-type *vgrR*. Therefore, whatever effect proteolysis has on *VgrS* does not appear to matter in terms of *VgrR* acting in a phosphorylation-dependent manner. Similar paradoxical behaviors are observed in other panels of this figure. This result casts doubts on the model of the paper.

Figure 7a-c: The authors need to repeat the EMSA assays with a non-specific DNA fragment because they use the specific competitor at very high levels.

Figure 7f and 7i: Please describe nature of *prc* mutation and location of *VgrR* binding in *prc* promoter and of primers used to examine *prc* RNA levels.

Supp Figure 2: What do you mean that "virulence levels were estimated by a semi-quantitative standard"?

Reviewer #2 (Remarks to the Author):

The mns by Deng et al. describes a novel mechanism of affecting signalling via a bacterial histidine kinase (HK) of a typical two-component system (2CS) by proteolysis of its sensor domain.

Overall, the data provided in this study are truly impressive in scope and quality and it reads almost too straightforward: All mutants/alleles always work, can be purified, are functional and everything always seems to work out and make sense... Despite the overall quality of the mns, I do have a number of comments and questions before this mns is ready for publication in Nat. Comm.

1. While the writing is mostly of good quality, some sections in the abstract and introduction do require polishing and some terms used there are rather ... unusual in their context, e.g. lines 18 (dominant receptors - they are not), line 19 (HK do not constitute "various 2CS"), lines 39/40 (beginning of this sentence), line 46 (term "mosiac-modular proteins" ???), lines 69/70. The very first two sentences of the introduction (lines 31-34) are particularly confusing.

2. Line 89: which DUF?

3. Lines 102-104: It is never explained why these particular phenotypes were selected and how they connect to the physiology of the Ptc-VgrRS cascade.

4. Line 108: I cannot see any particular cell envelope stress here.

5. Lines 126/127: How do these activities relate to other described Ser endopeptidases?

6. Lines 141/142: How do you know that this allele is still fully capable of binding its substrates?

7. Fig 3A as well as Fig. 6c seem rather pointless to me: what do I learn from these functional category plots?

8. The results on *VgrS* degradation by *Prc* puzzle me: if this peptidase only (and specifically) clips off 9 aa from the N-terminus of *VgrS* - how can it be that "*Prc* can completely degrade the *VgrS* sensor in vitro within 3 hours" (lines 194ff.) as documented in Fig. 4c? Likewise, for the

subsequent Western analysis, I am confused that clipping off approx. 20aa from VgrS does not lead to the appearance of an - at least slightly - smaller band in the lower panel (using the VgrS-specific antibody). Given that most of anti-HA signal has disappeared by 20 mins, the band in lane of the anti-VrS signal should turn into 2 bands to at least become "thicker" as a result of two different species of VgrS showing up together in one blot. Combined, the results shown in Figs 4c/4d raise quite a bit of concern.

9. Fig. 4h: What does "unrelated" mean in this context?? This is a clear signal that is not further explained anywhere. In fact, this band responds very specific, e.g. is highly unregulated in the del9 allele. How can this be unrelated/unspecific then?

10. Lines 305ff. I cannot quite follow the authors selection of putative VgrR target genes, particularly XC_0943. It does not encode any relevant function nor is it one of the highest scoring targets. The choice seems rather arbitrary.

11. Lines 336ff. Given the novelty and importance of their results, the authors do a rather poor job in discussing their findings. Large sections of this (too) short discussion are not very relevant since they refer again and again to unrelated systems or provide examples irrelevant for this study (lines 355-369, 386-392, 415-423). Instead, the most important questions are not addressed at all. That is, how is the proteolytic cleavage linked to stimulus perception both by the input domain of VgrS and - potentially - also by Prc? How does this combined processes affect autokinase activity? And how then does all of this relate to promoter binding and the activity of VgrR? The model in Fig. 8 does represent the results VERY insufficiently and does not explain much. Moreover, what is the advantage/physiological sense of adding proteolysis to the standard mode of stimulus perception by HKs?

Reviewer #3 (Remarks to the Author):

The manuscript entitled "Proteolysis of histidine kinase VgrS sequesters two-component signaling and promotes bacterial resistance to osmotic stress" by Deng et al. is an extremely thorough pursuit for a proteolytic substrate of the periplasmic protease Prc. It identified the TCS histidine kinase VgrS as a cleavage target, and showed that this cleavage affects the bacteria *Xanthomonas campestris* pv. *Campestris* (Xcc) in its response to osmotic and other stresses.

As a study of Prc and its regulation of VgrS per se, it is doubtlessly an outstanding tour de force. However, I have the following questions that need to be addressed.

1. It is unclear from the main text whether the cleavage of VgrS by Prc is generally significant. The authors acknowledge that "the irreversible modification of HKs in bacterial cells is less studied". Do their findings represent a new mode of regulation possibly applicable to many TCS systems, or is it a rather unique regulatory mechanism limited to the VgrS TCS in a small number of species? TCS is a common means by which bacteria sense environmental signals. One species could express "hundreds of HKs". How many of them could be regulated by periplasmic proteases?

1b. If the regulation of VgrS by Prc is restricted to Xcc and a few related species, then does this regulation impact a biological phenomenon important to us? In this case it would be the bacteria's virulence. The authors need to clearly communicate the significance of their findings, not just their technical prowess, which is evident from the results.

2. Here the authors' claim does not agree with their own data. They stated on Page 274 that the vgrS delta9 mutant decreased its virulence (Fig. 5e & 5f). However, Fig. 5e & 5f clearly showed that WT and the vgrS delta9 mutant have very similar levels of virulence.

2b. In addition, are the methods deployed in Fig. 5e & 5f common for examining virulence? Although outside the field, this reviewer has often seen the quantification of bacteria recovered from infected tissue as a standard way of measuring virulence. Is it not used for Xcc?

3. The data contain unexplained inconsistencies regarding the fate of VgrS sensor domain following cleavage. In vitro, Prc can not only cleave VgrS, but also degrade the whole sensor domain (Fig. 4c). However, in vivo the anti-VgrS band remained the same size, yet the antibody should detect a second band smaller by ~12 KDa (Fig. 4d). This difference needs to be reconciled.

3b. In Fig. 4c, why is the VgrS delta9 cleavage product not detected? It should be smaller than the uncleaved protein by about 1 KDa. One should be able to see a doublet.

4. The data in Fig. 4b cannot support the claim that "neither mature Prc nor inactive PrcS475A affected the autokinase activity of the truncated VgrS..." (Line 188), because this panel is missing VgrS autophosphorylation without Prc (WT or mutant). It is impossible to conclude whether Prc has an effect.

5. Why are the Kds of Prc-VgrS binding from SPR and MST so different? This should be explained for the uninitiated, such as this reviewer.

6. On the Coomassie of Fig. 3c, why do the two Prc bands have slightly different mobility? This reviewer assumes it is a denaturing gel.

Minor points

7. The two sentences between Lines 138 and 142 have incorrect grammar.

8. On Line 166, "To determine the region in the VgrS sensor that Prc binds" should be "To determine the region in Prc that binds the VgrS sensor", since the authors concluded "VgrS sensor binds the peptidase-DUF region of the Prc monomer".

9. On Line 172, the Kd is missing a unit.

Reviewers' comments:

Reviewer #1 (Remarks to the Author):

*The manuscript of Deng and co-workers reports a novel mechanism by which environmental signals control the activity of a two-component system. The authors argue that when the bacterium *X. campestris* experiences osmotic stress, the sensor protein VgrS is proteolysed by the protease Prc, which prevents VgrS autophosphorylation, therefore decreasing phosphorylation of VgrS cognate regulator VgrR, and abolishing transcription of VgrR-dependent genes. By contrast, in the absence of stress, Prc would not degrade VgrS resulting in VgrR phosphorylation and transcription of its target genes. Whereas proteases have been implicated in the stability of regulators belonging to the two-component system family, the proposed mechanism appears unique as the signal would, by virtue of proteolysing VgrS, render VgrS unable to conduct new rounds of autophosphorylation and phosphotransfer.*

The manuscript provides a lot of information. However, a lot of it is unclear or missing controls. In addition, the writing is not clear. There are too many spelling and grammatical errors to list them individually. In addition, the authors use words to mean things other than their accepted meanings.

Response 1-1. We thank the positive evaluation from the respected referee. According to the suggestion, we designed and performed the necessary experiments to address the questions as listed below. To improve the language, we carefully revised the manuscript and invited a commercial cooperation to help us to proofread the revision.

Below please find my specific comments.

Line 1: What do the authors mean by “sequesters”?

Response 1-2. The word “sequesters” was used to mean that proteolysis of the HK VgrS by Prc resulted in dephosphorylation of the VgrS-VgrR TCS. It is somewhat confused so that we changed the title into “...inhibits its autophosphorylation and...”. Please refer to the title (Line 1) in the revision.

Line 29: What do you mean by “effective mechanism”?

Response 1-3. The “effective mechanism” means that the proteolysis can block the protein phosphorylation of TCS. To make it clearer, the sentence is modified as “...HK proteolysis is a molecular mechanism to control the bacterial TCS signalling by modulating autokinase activity”. Please refer to Line 29 in the revision.

Line 48: The sentence starting with “Cellular regulation...” is not clear.

Response 1-4. Thanks, to make it clearer, the sentence is modified as “PDZ-domain containing proteases regulate physiological processes by binding or cleaving their protein substrates in cells”. Please refer to Line 48-49 in the revision.

Line 104: There is a discrepancy between the figure legend and the methods section. Did the authors use Fe2SO4 or Fe2Cl?

Response 1-5. We are sorry, this is a mistake in the legend of Fig. 1. It should be FeSO₄, rather than FeCl₂. We have corrected the same mistakes in the legends of Fig. 1 and 5.

Line 191: It is hard to follow the mutations on VgrS without a schematic of its predicted structure/conformation. The authors should refer to the location or amino acid region (as opposed to “sensor” domain).

Response 1-6. Thank to point out this problem. To improve it, we added schematic pictures of the secondary structure of VgrS in Fig. 3c in the revision. The cleavage site processed by Prc protease was shown in Fig. 5b in the revision. In addition, we revised the maintext to refer to the locations of the substituted amino acids in the VgrS.

Line 236: What is the autophosphorylation behavior of the A9G Q10A VgrS double mutant?

Response 1-7. According to the suggestion, we expressed and purified a recombinant VgrS^{A9G-Q10A} protein and determined its autokinase activity by *in vitro* phosphorylation assay. The result showed that the autophosphorylation level of VgrS^{A9G-Q10A} was not affected and similar to that of the WT VgrS. Although the the growth of the bacterial mutant $\Delta vgrS$ -vgrS^{A9G-Q10A} is similar to the WT in the NYG medium without osmostress, it grew slower than that of the WT strain under osmostress condition (1.0 M sorbitol), suggesting that substitutions of VgrS^{A9G-Q10A} residues are negative to bacterial survival to adapt the osmostress. Therefore, the facts that autophosphorylation level of VgrS^{A9G-Q10A} is not changed and VgrS^{A9G-Q10A} resists to the Prc cleavage provide biochemical evidences to explain that the substitutions around the Prc cleavage site (recombinant strain vgrS^{A9G-Q10A}) decreased the bacterial resistance to osmostress. We added the results of autophosphorylation of VgrS^{A9G-Q10A} in the Fig. 5g and the growth curves of vgrS^{A9G-Q10A} strain in Fig. 6c and 6d of revision. Related descriptions were revised, as shown in the Lines 266-269 of the revision.

Line 240: Contrary to what the authors state, amino acids 2 to 9 are not essential for autokinase activity, otherwise there would be no VgrS phosphorylation.

Response 1-8. We agree to the comment. The description was the revised. Please refer to Line 266-269 in the revision.

Line 249: The authors should provide growth data on media where there is no stress.

Response 1-9. Sure, according to the suggestion, we measured the bacterial growth under the unstimulated condition (NYG medium), the result showed that there is no remarkable difference between the WT strain and the *vgrS*^{Δ9}. We added the data in the Fig. 6 and described the result in Lines 278-279 in the revision.

Line 278: The authors did not show dephosphorylation of VgrR.

Response 1-10. Sure, according to the comment, we revised the description as "...phosphorylation of the VgrS is detrimental to the bacterial growth...". Please refer to Line 306 in the revision.

Line 297: There are other interpretations for the findings.

Response 1-11. Sure, we revised the description as: "...a possible reason is that *prc* regulates the VgrR-promoter binding so that *prc* deletion causes change in the binding events". Please refer to Line 324-326 in the revision.

Line 321: Are the authors are really measuring the effects of three genes? They are examining the effect of prc inactivation and one of the genes being investigated is prc itself. Please explain.

Response 1-12. Sure, because we constructed a non-polar, in-frame deletion mutant of *prc* by deleting the coding sequence from 223 to 713 aa., we used qRT-PCR assay to measure the mRNA level of the truncated *prc* gene. *prc* promoter has a VgrR-binding motif and its transcription is subject to the control of VgrR.

To genetically investigate the roles of these candidate genes in stress response, we constructed two in-frame deletion mutants of the *XC_0943* and *XC_2164*. In addition, we also constructed other four in-frame deletion mutants of the downstream, candidate genes of *XC_0690* (encoding a sugar kinase), *XC_3300* (outer-membrane protein), *XC_3301* (oxidoreductase) and *XC_3576* (outer-membrane protein). Although the deletion of *XC_0690* and *XC_3576* did not impact the bacterial resistance to osmostress (Fig. 8a in the revision), deletion of the other four genes, including *XC_0943*, *XC_2164*, *XC_3300* and *XC_3301*, resulted in significant decrease of bacterial growth under osmostress condition (Fig. 8a in the revision). As EMSA and qRT-PCR analyses revealed, these four genes are subject to the control of VgrR since directly VgrR binds to their promoter and positively regulates their transcription levels. In the *prc* mutant, their mRNA levels were also significantly decreased. Therefore, these results demonstrate that *XC_0943*, *XC_2164*, *XC_3300* and *XC_3301* are osmostress response genes that are controlled by VgrR and Prc. These genetic results were put into Fig. 8 and Supplementary Fig. 8, and the related descriptions were revised. Please refer to Line 343-351 in the revision.

Please refer to the response to the second reviewer (Response 2-11) for the similar suggestion.

Line 339: Please provide original citation for example of a given species encoding "hundreds of HKs".

Response 1-13. Sure, the phrase was modified into "...bacterial usually encode several to over a hundred of HKs.....", and we cite two references in which mentioned some bacteria encode a large number of HKs (Ortet et al, 2014. *Nucleic Acids Res.* 43:D536-D541; Shi et al, 2008. *J Bacteriol*, 190:613-524). For example, *Mycococcus* sp. encodes approximately 130-150 HKs. Please refer to Lines 380-381 for the revision.

Figure 4c: Degradation appears very inefficient.

Response 1-14. Yes, Prc can promptly degrade the general substrate of β -casein *in vitro* (Fig. 2a in revision) and cleave VgrS sensor *in vivo* (< 5 min, Fig. 4e in revision), it cleaved and degraded the VgrS sensor within 60 min *in vitro*. It has been reported previously that when cleaving physiological substrates *in vitro*, bacterial PDZ domain-containing protease exhibited low activity since additional adaptor proteins or other factors are needed to facilitate the degradation (Battesti and Gottesman, 2013. *Current Opin Microbiol.* 16:140-147; Su et al, 2018. *Nat Commun.* 8:1516.). Comparison of the degrading dynamics in cleaving β -casein and VgrS sensor suggests the existence of adaptor proteins of Prc. According to this, we described the difference and discussed the possible reason. Our future work will focus on the activation mechanism of Prc protease and additional cellular factors that impact the activity. Please refer to Line 459-471 in the discussion part.

Figure 5b: Differences are small.

Response 1-15. This experiment was replicated 3 times, and the results were very stable. The relative survival rates of the *vgrS*^{A9} mutant were 1.33 times the WT strain, and statistical analysis indicated that the differences are significant. This result also suggests that deletion of the N-terminal amino acids of VgrS promoted the bacterial resistance to osmopressure. In the revision, we used growth curve of bacteria strains under the stress or no stress condition to show the phenotypic change. The data was collected by an automatic machine and was very stable. We turn to use the growth curves to describe the bacterial adaptation under the osmopressure. Please refer to Fig. 6 and related descriptions (Line 278-284 in revision) for details.

Figure 5c: The vgrS and vgrR mutants behave differently from each other. Therefore, it is not clear what is really going on in terms of the relationship between phenotype and genotype. Moreover, the vgrR mutant specifying the D51A variant, which is predicted to not be subject to phosphorylation complements a $\Delta vgrR$ strain just like the wild-type vgrR. Therefore, whatever effect proteolysis has on VgrS does not appear to matter in terms of VgrR acting in a phosphorylation-dependent manner. Similar paradoxical behaviors are observed in other panels of this figure. This result casts doubts on the model of the paper.

Response 1-16. Our data showed that various mutations in *vgrS* or *vgrR*^{D51A} substitution promoted the bacterial resistance to osmopressure since the mutation of *vgrS* or *vgrR*^{D51A} led to low level of VgrR phosphorylation (Fig. 6 in revision). These results are consistent. Genetically deletion of *vgrR* is negative because there is no VgrR (both phosphorylated or dephosphorylated) anymore in bacterial cells. Deletion of the *vgrR* significantly decreased the bacterial growth under osmopressure condition (Fig. 6f in revision), suggesting that the dephosphorylated VgrR, rather a null mutant, is needed to bacterial resistance to osmopressure.

In addition, in the study of TCS regulation, it is generally found that the phenotypic change of HK gene deletion is weaker than that of the RR gene deletion, possibly caused by that the small molecular phosphate donors, such as acetyl phosphate, could phosphorylate RR.

In the previous version, we exhibited the data of bacterial growth by showing the colonies on the plate (NYG plus 1.0 M sorbitol). This data may cause confusion because it is not quantitative and hardly to be compared if the difference is not so obvious. To make improvement, we measured the growth curves of bacterial strains under the stress condition. As shown in Fig. 6 of revision, while the growth of *vgrR* deletion mutant showed remarkably decrease in NYG-sorbitol medium, the growth of *vgrR*^{D51A} is quicker than that of the WT strain, suggesting that constitutively dephosphorylation of VgrR is positive for bacterial stress response. Please refer to Lines 287-293 in the revision.

Similarly, the growth curves of bacterial strains showed that the resistance of both *vgrS* deletion mutant and *vgrS*^{H186A} mutant (containing constitutively dephosphorylated VgrS) were higher than that of the WT strain, while the recombinant *vgrS*^{A9G-Q10A} strain grew slower than the WT strain. The result of the growth curve is clearer than the previous bacterial colony morphology or relative survival rate, and we use this set of data in the revision. Please refer to Fig. 6 and Line 278-284 in the revision.

Figure 7a-c: The authors need to repeat the EMSA assays with a non-specific DNA fragment because they use the specific competitor at very high levels.

Response 1-17. Yes, when performing the EMSA assay, high concentration of poly(dI::dC) has been added in the reaction mixture to suppress the non-specific binding between VgrR and the DNA probes, this ensured that the signal of the specific protein-DNA binding was observed. According to this suggestion, we further used non-specific competitor in the EMSA assay. The non-specific competitor did not decrease the intensity of isotopic signals that represent the VgrR-promoter binding. These results demonstrate that VgrR specifically bind to the promoter regions of downstream genes. We added the experimental data in the Fig. 7d, Fig. 8b, 8c and Supplementary Fig. 2a, 2b in revision. Please refer to these figures and their legends for a detail.

*Figure 7f and 7i: Please describe nature of *prc* mutation and location of VgrR binding in *prc* promoter and of primers used to examine *prc* RNA levels.*

Response 1-18. Sure, according to this comment, we added the sequences containing the VgrR-binding motif to each of the tested promoters. In addition, we added description on how to examine *prc* RNA level in the maintext. Because we in-frame deleted the DNA sequence from 669th-2139th (coding the 223-713 aa) of *prc*, the *prc* mRNA level was detected by measuring the remaining, truncated *prc* mRNA (497-665 nt). In addition, the binding sequences and motifs were showed in each of the EMSA results. Please refer to Fig. 7d, Fig. 8b, 8c and Supplementary Fig. 8a and 8b in the revision.

Supp Figure 2: What do you mean that “virulence levels where estimated by a semi-quantitative standard”?

Response 1-19. Yes, because the shape of cruciferous plant leaves not long and thin, it is difficult to measure the lesion length. When measuring the virulence level of black rot disease,

our colleagues usually estimate the virulence scores by the symptoms of the disease (0 ~ 4 levels, Dow et al, 1990. *Appl Environ Microbiol*, 56:2994-2998). This semi-quantitative standard was also adopted in our previous studies. According to this comment, we added the standard of virulence scores in the Materials and Methods, please refer to Lines 539-546 in the revision.

Reviewer #2 (Remarks to the Author):

The mns by Deng et al. describes a novel mechanism of affecting signalling via a bacterial histidine kinase (HK) of a typical two-component system (2CS) by proteolysis of its sensor domain.

Overall, the data provided in this study are truly impressive in scope and quality and it reads almost too straightforward: All mutants/alleles always work, can be purified, are functional and everything always seems to work out and make sense... Despite the overall quality of the mns, I do have a number of comments and questions before this mns is ready for publication in Nat. Comm.

Response 2-1. We acknowledge the positive evaluation and encouragement from the respected referee. This work is not smooth since the Prc of *X. oryzae* pv. *oryzae*, which was studied when the project was initiated eight years ago, is hardly to be purified. After we turn to study the Prc of *X. campestris* pv. *campestris*, the situation was improved since the enzymatic activity of the protease is detectable and can be biochemically investigated. According to the suggestion listed below, we performed new experiments and modified the maintext to answer the questions.

1. While the writing is mostly of good quality, some sections in the abstract and introduction do require polishing and some terms used there are rather ... unusual in their context, e.g. lines 18 (dominant receptors - they are not), line 19 (HK do not constitute "various 2CS"), lines 39/40 (beginning of this sentence), line 46 (term "mosaic-modular proteins" ???), lines 69/70. The very first two sentences of the introduction (lines 31-34) are particularly confusing.

Response 2-2. Many thanks, according to the comment, we carefully checked the manuscript and revised the language. A commercial cooperation was also invited to help us to proofread the text.

Line 18, "dominant receptors" was modified as "extracytoplasmic receptors";

Line 19, revised as "HK and its cognate response regulator constitute a two-component signalling system...".

Line 31-34. Revised. Please refer to Line 31-32 in the revision.

Line 39-40. Revised as "...how proteolysis modifies the receptor kinases and controls their regulatory functions remains to be an unanswered question". Please refer to Line 39-40 in the revision.

Line 46. "mosaic-modular proteins" was changed to "multiple-domain containing proteins".

Line 69/70. Revised. Please refer to Line 67-68 in the revision.

2. Line 89: which DUF?

Response 2-3. Sure, the information was added, the DUF domain of Prc is DUF3340. Please refer to Line 85 in the revision.

3. Lines 102-104: It is never explained why these particular phenotypes were selected and how they connect to the physiology of the Prc-VgsRS cascade.

Response 2-4. Sure. In fact, previously we performed extensive phenotypic characterization of the *prc* mutant, including biofilm formation, extracellular enzyme activity, production of exocellular polysaccharides, resistance to various environmental stresses (such as metals, oxidative stress, osmolarity, pH, temperature), drug resistance, etc (the following Fig. 1R shows an example). Only the phenotypes with substantial changes were selected to be reported. As our results indicated, Prc specifically cleavages VgrS when the bacterium is challenged by osmstress. Under other tested environmental stresses, it seems Prc did not directly modulate the VgrS-VgrR signalling. According to the comment, we added the rationale for the selection of these phenotypes in the study. Please refer to Line 96-99 in the revision.

Fig. 1R. Phenotypic characterization of the *prc* mutant

4. Line 108: I cannot see any particular cell envelope stress here.

Response 2-5. Sure, we agree to the suggestion and revised the sentence as “...involved in bacterial virulence, resistance to antibiotics, metal and osmolarity stresses”. Please refer to Line 180 in the revision.

5. Lines 126/127: How do these activities relate to other described Ser endopeptidases?

Response 2-6. According to the comment, we compared the the activity of Prc activity and other reported Ser endopeptidases. It showed that Prc has a relative strong protease activity since the HtrA or Tsp proteases usually have K_m value from micromol to minimol levels. The K_m value of Prc/Tsp of *E coli* is $4.4 \pm 0.6 \mu\text{M}$ (Beebe et al, 2000. *Biochemistry*).

39:3149-3155). We added this comparison and the related references in the description, please refer to Line 128-130 in the revision.

6. *Lines 141/142: How do you know that this allele is still fully capable of binding its substrates?*

Response 2-7. Thanks for the comment. Because the active form of protease will promptly degrade its substrate and release them, to screen for the binding proteins of a protease, generally an inactive, recombinant protease has to be used [Lopez-Otin et al, 2002. *Nat Rev Mol Cell Biol.* 3:509-519; Overall et al, 2007. *Nat Rev Mol Cell Biol.* 8:245-257]. In this study, we substituted the Ser⁴⁷⁵ residue by Ala and used the recombinant strain containing this *prc* allele (Δ prc-prc^{S475A}-HA-FLAG) in the affinity proteomic analysis. This Prc^{S475A} is inactive so that the recombinant strain exhibited significantly decreased resistance to osmopressure (as the following figure shown). To address this comment, we then constructed a recombinant strain (Δ prc-prc-HA-FLAG) containing the WT *prc* with C-terminal HA-FLAG tags. Under the osmopressure condition, the stress resistance of this strain showed only slight decrease when compared to the WT strain (following Fig. 2R and Supplementary Fig. 2f in revision), suggesting that the capability in binding substrates of this recombinant Prc is largely maintained. In addition, previous studies and our data revealed that the C-terminal of tail specific protease (including Prc) is not involved in the substrate recognition (Su et al, 2018. *Nat Commun.* 8:1516). We described this result in Line 145-148 of the revision.

Fig. 2R. Resistance to osmopressure of bacterial strains. Strains were grown on the NYG-Sorbitol plate for six days. Note that *prc* deletion mutant and Δ prc-prc^{S475A}-HA-FLAG were highly susceptible to osmopressure, while the Δ prc-prc-HA-FLAG strain grew similar to that of the WT strain.

7. *Fig 3A as well as Fig. 6c seem rather pointless to me: what do I learn from these functional category plots?*

Response 2-8. Fig. 3a is the functional category of the potential Prc-binding proteins obtained by the affinity proteomic study (TAP analysis), while previous Fig. 6c (Fig. 7a in revision) is the comparison of the VgrR-binding promoters between the WT and *prc* mutants. These two figures provided the genome-scale information about the Prc-binding proteins and the impact of *prc* mutation in the VgrR-binding affinity. They are the basis for in-depth functional verification and investigation. For this reason, we suggest to keep them in the revision. In addition, we revised the description of the two results in the maintext to improve them. Please refer to Line 155-157 and Line 323-342 in the revision.

8. *The results on VgrS degradation by Prc puzzle me: if this peptidase only (and specifically) clips off 9 aa from the N-terminus of VgrS - how can it be that "Prc can completely degrade the VgrS sensor in vitro within 3 hours" (lines 194ff.) as documented in Fig. 4c? Likewise, for*

the subsequent Western analysis, I am confused that clipping off approx. 20aa from VgrS does not lead to the appearance of an - at least slightly - smaller band in the lower panel (using the VgrS-specific antibody). Given that most of anti-HA signal has disappeared by 20 mins, the band in lane of the anti-VgrS signal should turn into 2 bands to at least become "thicker" as a result of two different species of VgrS showing up together in one blot. Combined, the results shown in Figs 4c/4d raise quite a bit of concern.

Response 2-9. Thanks for this comment. Our previous description about Prc protease activity is somewhat confusing. Prc belongs to the tail-specific protease (Tsp) family. This kind of proteases degrade substrate by two ways: they can process the specific proteins by removing their C termini, such as FtsI and penicillin-binding protein 3 of *E. coli*, in addition, they can also gradually degrades protein substrates into small peptides in a long-time treatment. For example, Prc of *E. coli* could completely degrade an outer membrane lipoprotein MepS *in vitro* within 120 min that is involved in peptidoglycan synthesis (Beebe et al, 2000. *Biochemistry*. 39:3149-3155; Su et al. *Nat Commun*. 8:1516.). The results of previous Fig. 4c indicated that Prc gradually cleaves and degrades VgrS sensor. To make an improvement, we decreased the temperature of proteolysis from 28°C to 20°C to slow down the reaction, and used silver-staining to detect the proteolytic products. It showed that VgrS sensor was gradually degraded by Prc (Fig. 4c and 4d in revision).

After cleaving by the Prc protease between the A9-Q10 site, the theoretical molecular weight of VgrS is changed from 43.76 kD to 42.72 kD. However, in the SDS-PAGE gel, this difference (1 kD) is too small to be detected, especially considering that the molecular weight is between the 32 kD to 56 kD range in the SDS-PAGE gel (as shown in the following Fig. R3). We tried to optimize various conditions of electrophoresis but the results revealed that traditional SDS-PAGE is not fit to detect the small difference. After discussing this problem with our colleagues who are experts in protein science and structural biology (Prof. Yong Tao and Prof. George Fu Gao in the IM-CAS), we tried our best to detect the cleaved products by several strategies. Most of them did not work, but eventually the mass spectrometry (MS) analysis directly detected the existence of the cleaved, N-terminal peptide by Prc proteolysis. We briefly summarize our efforts here:

Fig. 3R. SDS-PAGE cannot separate the full-length VgrS and truncated VgrS with the first eight aa. being deleted. Inverted membrane vesicles of VgrS were separated by 12% SDS-PAGE and detected by Western blotting.

First, we tried to add three or four tandemly repeated arginines (Arg) into the N-terminal of VgrS (after the first methionine) since that existence of alkaline amino acid residues (especially Arg) in the peptide could remarkably change the isoelectric point of the protein and then cause difference in the mobility of recombinant protein in the electrophoresis. If the VgrS sensor containing these Arg residues were removed by Prc proteolysis, the cleaved product will be separated from the intact VgrS. However, although we successfully obtained the recombinant VgrS containing three or four tandemly repeated Arg residues, functional Prc

treatment did not change the autophosphorylation levels of these recombinant VgrS (as shown in the following Fig. 4R-a), suggesting that Prc did not cleave them anymore (note that inactive Prc^{S475A} addition even increased the VgrS phosphorylation level, possibly caused by stabilizing VgrS via protein-protein interaction). This is most probably caused by that the introduction of tandemly-repeated Arg affected the conformation of the sensor and interfered with Prc recognition. 2). Similar to this strategy, we also obtained a recombinant VgrS fused with 3 × FLAG tags in the N-terminal. Prc also did not affect the autokinase activity of this 3 × FLAG-VgrS (Fig. 4R-b), indicating that the addition of 24 aa-length, 3 × FLAG tags in full-length VgrS or VgrS sensor is detrimental to the proteolysis. Resistance to the Prc proteolysis of these recombinant VgrS were also confirmed by the result of Fig. 4R-c, in which the recombinant VgrS with tandemly repeated Args or 3 × FLAG tags were treated by functional Prc protease. 3). In addition, we also constructed a recombinant strain containing 3 × FLAG tag fused at the N-terminal of VgrS (strain ΔvgrS-3*Flag-vgrS), as shown in Fig. 4R-d, the fusion of the 3 × FLAG tag made the VgrS unstable since two major bands were detected by Western blotting. 4). Furthermore, we tried to use N-terminal sequencing to detect the cleavage events on VgrS sensor. But this method is also not work because that the full-length VgrS is embedded in inverted membrane vesicles that containing other proteins (the abundance of VgrS is about 50%), while the N-terminal sequencing requires a high purity proteins (> 95%) to be analyzed.

Fig. 4R. Various recombinant VgrS are resistance to Prc proteolysis. **a.** The autokinase activities of VgrS containing tandemly ranged Arg residues were blind to the treatment of Prc. **b.** The autokinase activity of VgrS with N-terminal, 3 × FLAG tag was unaffected by the Prc treatment. **c.** Prc did not cleaves the recombinant VgrS containing tandemly ranged Arg residues and VgrS with N-terminal, 3 × FLAG tag. **d.** Bacterial strain containing a recombinant 3 × FLAG tag VgrS is unstable.

To solve the above problem, we turn to directly identify the cleaved N-terminal product of VgrS from multiple proteins. A peptide of VgrS sensor (2nd to 9th aa. NRNIDFFA) was chemically synthesized and used as a standard in the QTRAP-LC-MS/MS analysis. The results showed that after 5 min of Prc protease treatment, this peptide can be detected by the MS analysis (Supplementary Fig. 6 in revision). Along with the processing time (5, 10, 30, 60,

120, 180 min), the amounts of this peptide were promptly accumulated, suggesting that Prc cleaves the VgrS at the A9-Q10 site. In our previous study, we have employed MALDI-TOF-MS/MS analysis to show that the purified VgrS sensor was cleaved at this site (< 10 min, Supplementary Fig. 5). Collectively, these results demonstrate that the A9-Q10 of VgrS is the primary cleavage site of Prc protease.

Based on the results of the afore-mentioned experiments, we added the experimental data in revision (Fig. 5e and Supplementary Fig. 6) and revised the related descriptions in the maintext. Please refer to Line 247-257 in the revision.

9. Fig. 4h: What does "unrelated" mean in this context? This is a clear signal that is not further explained anywhere. In fact, this band responds very specific, e.g. is highly upregulated in the *del9* allele. How can this be unrelated/unspecific then?

Response 2-10. Sure, our repeated experiments showed that this unrelated band is stable and is a miscellaneous signal (the following Fig. 5R). It is not the VgrS-P signal because the position of this band is not right (as indicated in the result of Western blotting in the lower panel of Fig. 5f in revision). Because the full-length VgrS is embedded in the inverted membrane vesicle, this isotopic signal might be caused by other proteins that were phosphorylated in the reaction mixture.

Fig. 5R. Deletion of the N-terminal eight residues of VgrS decreases its autophosphorylation level.

10. Lines 305ff. I cannot quite follow the authors selection of putative VgrR target genes, particularly *XC_0943*. It does not encode any relevant function nor is it one of the highest scoring targets. The choice seems rather arbitrary.

Response 2-11. Yes, we agree with this comment. Because there lack the genome-scale or mechanistic studies on the osmopressure resistance of *X. campestris* pv. *campestris*, we chose two genes regulated by VgrR, *XC_0943* and *XC_2164* for further analysis. *XC_0943* is a function-unknown gene, while the ortholog of *XC_2164* (*yciE*) were identified to be involved in the osmopressure response of *E. coli*. The roles of *XC_0943* and *XC_2164* in stress response are undetermined previously.

To make an improvement, besides *XC_0943* and *XC_2164*, we selected other four candidate genes from the result of ChIP-seq to investigate their roles in osmopressure tolerance, including *XC_0690* (encoding a sugar kinase), *XC_3300* (outer membrane protein), *XC_3301* (oxidoreductase) and *XC_3576* (outer-membrane protein) that may be involved in osmopressure response. We constructed in-frame deletion mutants of the six genes. Inactivation of *XC_0943*, *XC_2164*, *XC_3300* and *XC_3301* resulted in remarkable decrease of bacterial growths on the

NYG plate plus 1.0 M sorbitol (Fig. 8a in revision), and the transcriptional levels of the four genes were induced upon the sorbitol stimulation (Fig. 8 and Supplementary Fig. 8 in revision). These results suggest that the four genes are participated in stress response. Inactivation of *XC_0690* and *XC_3576* did not have impact on the bacterial growth.

In addition, EMSA together with qRT-PCR assay demonstrated that VgrR directly bound to the promoter regions of these four genes and positively regulated their transcriptions, since the mRNA levels of them significantly decreased in the *vgrR* mutant, as well as in the *prc* mutant (Fig. 8 and Supplementary Fig. 8 in revision). Collectively, these analyses confirmed that *XC_0943* and *XC_2164*, *XC_3300* and *XC_3301* are osmstress-response genes that are subject to the control of Prc-VgrS-VgrR cascade. Based on the results, we revised the manuscript and added the above data in Fig. 8 and Supplementary Fig. 8. Please refer to Line 343-351 in the revision.

A similar question was also raised by the first referee, please refer to the response (Response 1-12) for the information.

11. Lines 336ff. Given the novelty and importance of their results, the authors do a rather poor job in discussing their findings. Large sections of this (too) short discussion are not very relevant since they refer again and again to unrelated systems or provide examples irrelevant for this study (lines 355-369, 386-392, 415-423). Instead, the most important questions are not addressed at all. That is, how is the proteolytic cleavage linked to stimulus perception both by the input domain of VgrS and - potentially - also by Prc? How does this combined processes affect autokinase activity? And how then does all of this relate to promoter binding and the activity of VgrR? The model in Fig. 8 does represent the results VERY insufficiently and does not explain much. Moreover, what is the advantage/physiological sense of adding proteolysis to the standard mode of stimulus perception by HKs?

Response 2-12. We thank this suggestion to improve our discussion. According to the comment, we re-wrote the discussion part to extensively focus on the Prc regulation and its relationship with VgrS. The revised discussion mainly focused on the following points as suggested:

- 1) The meaning of proteolysis of HKs in regulating the bacterial adaptation.
 - 2) The advantages of proteolysis in regulating phosphorylation process of TCS.
 - 3) The role of VgrS-VgrR in regulating virulence and stress response and the relationship of the TCS and Prc proteolysis.
 - 4) The possibility of the involvement of adaptor proteins in Prc proteolysis.
- Please refer to the revised Discussion for the details.

Reviewer #3 (Remarks to the Author):

*The manuscript entitled “Proteolysis of histidine kinase VgrS sequesters two-component signaling and promotes bacterial resistance to osmstress” by Deng et al. is an extremely thorough pursuit for a proteolytic substrate of the periplasmic protease Prc. It identified the TCS histidine kinase VgrS as a cleavage target, and showed that this cleavage affects the bacteria *Xanthomonas campestris* pv. *campestris* (*Xcc*) in its response to osmotic and other*

stresses.

As a study of Prc and its regulation of VgrS per se, it is doubtlessly an outstanding tour de force. However, I have the following questions that need to be addressed.

Response 3-1. We thank the referee for the encouragement and comments. According to the suggestion, we thoroughly revised the manuscript and collected necessary data to address the question listed below. Please refer to the responses for details.

1. It is unclear from the main text whether the cleavage of VgrS by Prc is generally significant. The authors acknowledge that “the irreversible modification of HKs in bacterial cells is less studied”. Do their findings represent a new mode of regulation possibly applicable to many TCS systems, or is it a rather unique regulatory mechanism limited to the VgrS TCS in a small number of species? TCS is a common means by which bacteria sense environmental signals. One species could express “hundreds of HKs”. How many of them could be regulated by periplasmic proteases?

1b. If the regulation of VgrS by Prc is restricted to Xcc and a few related species, then does this regulation impact a biological phenomenon important to us? In this case it would be the bacteria’s virulence. The authors need to clearly communicate the significance of their findings, not just their technical prowess, which is evident from the results.

Response 3-2. Response 3-2. Thanks to the suggestion. Actually Prc/Tsp is a periplasmic, PDZ-domain containing protease widely distributed in the Gram-negative bacteria, and most histidine kinases have periplasmic sensors to detect environmental stimuli. Although Prc-HK interaction may exist in other bacteria and play important roles in bacterial adaptation, it is somewhat hard to infer the notion now because 1) There is no consensus, recognition/cleavage sequence of the Prc/Tsp family proteases so that it is quite hard to determine the interactions between these proteases and HKs by bioinformatics approach; 2) As shown in Supplementary Table 3, our affinity proteomic analysis (TAP) just identified a HK (VgrS), albeit *X. campestris* pv. *campestris* encodes a large number of HKs (52 proteins); 3) As our results suggested, the cleavage of VgrS sensor by Prc proteolysis is specific to the osmotic stress response. When the bacterium is challenged by other stresses, including virulence, metal stress and antibiotic resistance, it seems that Prc did not have regulatory relationship with VgrS (Supplementary Fig. 7 in revision) during these physiological processes. Based on these results, we tend to believe that proteolysis of HK is a newly identified (or even important) mechanism to modulate the TCS signaling, but further case-by-case study is needed to investigate the significance of this kind of pathway in other bacteria. According to the suggestion, we revised and reinforced the related discussion, please refer to Lines 398-423 in the revision.

2. Here the authors’ claim does not agree with their own data. They stated on Page 274 that the vgrS delta9 mutant decreased its virulence (Fig. 5e & 5f). However, Fig. 5e & 5f clearly showed that WT and the vgrS delta9 mutant have very similar levels of virulence.

Response 3-3. Sure, previous Fig. 5e and 5f and our repeated experiments showed that the virulence level of the strain *vgrS*^{Δ9} was just slightly decreased (not “significantly decreased”) when compared to that of the WT strain. This is in parallel to our previous study that found the deletion of *vgrS* caused slight decrease in virulence (Wang et al, 2016. PLoS

Pathogens, 12:e1006133). We have corrected this descriptive mistake. Please refer to Line 302-303 in the revision.

2b. In addition, are the methods deployed in Fig. 5e & 5f common for examining virulence? Although outside the field, this reviewer has often seen the quantification of bacteria recovered from infected tissue as a standard way of measuring virulence. Is it not used for Xcc?

Response 3-4. Sure, according to the suggestion, we performed *in-planta growth assay* to measure the dynamics of bacterial populations after inoculation into the host plant leaves. The results were added in the Supplementary Fig. 2c and Supplementary Fig. 7c. The related descriptions were also modified, please refer to Lines 101 and 302 in the revision.

3. The data contain unexplained inconsistencies regarding the fate of VgrS sensor domain following cleavage. In vitro, Prc can not only cleave VgrS, but also degrade the whole sensor domain (Fig. 4c). However, in vivo the anti-VgrS band remained the same size, yet the antibody should detect a second band smaller by ~12 KDa (Fig. 4d). This difference needs to be reconciled.

3b. In Fig. 4c, why is the VgrS delta9 cleavage product not detected? It should be smaller than the uncleaved protein by about 1 KDa. One should be able to see a doublet.

Response 3-5. Thanks for the suggestion. This comment is similar to the question raised by the second reviewer. Please refer to the Response 2-9 for a more detailed analysis and experiments.

Prc/Tsp-family proteases not only have the endopeptidase activities to cleave the protein substrate, especially from the C-termini, but also have the peptidase activities to completely degrade the substrates into small peptides in the extended time (Beebe et al, 2000. *Biochemistry*. 39:3149-3155; Su et al. *Nat Commun*. 8:1516). To make an improvement, we decrease the temperature of proteolysis from 28°C to 20°C and used silver staining to detect the proteolytic products of VgrS. As shown in Fig. 4c and 4d of revision, Prc cleaved and gradually degraded the purified VgrS sensor *in vitro*.

After Prc cleavage, the theoretical molecular weight of VgrS is changed from 43.76 kD to 42.72 kD. As shown in Fig. 3R in this reply, ordinary SDS-PAGE gel is hard to separate such as small difference (nearly 1 kD) in the range between 32 kD to 56 kD. Therefore, we used several approaches to separate the full-length VgrS and the truncated VgrS, including addition of 3-4 tandemly repeated Arg residues in the N-terminal of VgrS to change its behavior in electrophoresis, or fusion of a 3 × FLAG tag in the N-terminal of VgrS. However, subsequent analysis revealed that these recombinant VgrS resisted to the Prc proteolysis and their autophosphorylation level were unaffected by Prc treatment (as shown in Fig. 4R in reply). In addition, in a recombinant strain encoding a 3 × FLAG-VgrS, the VgrS is unstable (Fig. 4R-c). N-terminal sequencing of the proteolytic products was not working because the full-length VgrS was embedded in an inverted membrane vesicles containing other proteins, while the N-terminal sequencing requires a highly purified protein (> 95%).

Eventually, we turn to directly detect the cleaved, N-terminal peptide by high-resolution mass spectrum. A short peptide (2nd to 9th aa. of VgrS, NRNIDFFA) was chemically synthesized and used as a standard in the MS analysis. QTRAP-LC-MS/MS analysis

successfully detected this peptide after 5 min of Prc treatment, and the amounts of the peptide gradually increased along with the processing time. Therefore, this experiment demonstrated that when the VgrS IMV was co-incubated with Prc, Prc primarily cleaved the VgrS sensor between the A9-Q10 site. Based on these results, we reported the experimental data in the revision (Fig. 5e and Supplementary Fig. 6) and revised the related descriptions in the maintext. Please refer to Line 247-257 in the revision.

4. The data in Fig. 4b cannot support the claim that “neither mature Prc nor inactive PrcS475A affected the autokinase activity of the truncated VgrS...” (Line 188), because this panel is missing VgrS autophosphorylation without Prc (WT or mutant). it is impossible to conclude whether Prc has an effect.

Response 3-6. Thank you very much to reminder us. There is a labelling mistake on the upper panel of Fig. 4b: the first column of the row of “Prc” should be “-” (no addition), rather than “+” (addition). The time course of reaction of addition of both Prc and Prc^{S475A} were 0.5, 1, 5, 5 and 10 min. Please note that there is no “Prc +” before the addition of Prc^{S475A}. The intensity of the isotopic signal of this band is similar to the other hands. We have corrected this mistake in the revision (Fig. 4b, the first lane).

5. Why are the Kds of Prc-VgrS binding from SPR and MST so different? This should be explained for the uninitiated, such as this reviewer.

Response 3-7. Yes, although both SPR and MST assays revealed that Prc physically bound to VgrS sensor *in vitro*, the K_d values from the two methods were different. The Prc-VgrS sensor binding affinity detected by SPR is 33.9 μM , while K_d value measured by MST is 0.65 μM , much stronger than that of the SPR. This is caused by the difference of the two technologies. Since Prc-VgrS sensor interaction was detected on a solid, CM5 sensor chip to embed Prc during SPR analysis (flowed in a PBST solution which contains 0.005% Tween-20), while this binding event was directly detected in a solution by MST (in a NTA buffer), we tend to believe that the result from MST is close to reality. Because the SPR data is complete and the MST was used to verify important interactions, we selected to report the SPR data in the figures and report the MST data in the supplementary figures. According to the comment, we added an explanation on the difference between the SPR and MST analyses. Please refer to Line 177-180 in the revision.

6. On the Coomassie of Fig. 3c, why do the two Prc bands have slightly different mobility? This reviewer assumes it is a denaturing gel.

Response 3-8. Yes, Fig. 3c used a SDS-PAGE gel to detect the purified Prc^{S475A}. The difference was caused by the electrophoresis since other repeats did not have this phenomenon. The Western blotting in the lower panel also showed that the two bands were in the same position. We changed this picture by another replicate. Please refer to Fig. 3c in revision.

Minor points

7. The two sentences between Lines 138 and 142 have incorrect grammar.

Response 3-9. Thanks, the sentence is revised as: “...a recombinant bacterial strain

(Δ prc-prc^{S475A}-HA-FLAG) was constructed. In the strain, an HA and a FLAG epitope tags were tandemly fused to the C-terminal of an inactive Prc whose Ser⁴⁷⁵ was substituted by Ala". Please refer to Lines 142-144 in the revision.

8. *On Line 166, "To determine the region in the VgrS sensor that Prc binds" should be "To determine the region in Prc that binds the VgrS sensor", since the authors concluded "VgrS sensor binds the peptidase–DUF region of the Prc monomer".*

Response 3-10. Done, the comment is correct, the previous description incorrectly refers to the VgrS sensor region, it should be Prc region. Please refer to Lines 181-182 in the revision.

9. *On Line 172, the Kd is missing a unit.*

Response 3-11: Thank you to point out this mistake, the unit "μ" should be "μM", we have corrected this word. Please refer to Lines 187 in the revision.

Reviewers' comments:

Reviewer #1 (Remarks to the Author):

The revised manuscript by Deng and colleagues provides answers to many of the questions that were raised by the three reviewers of the original submission. Although the authors indicate that they "invited a commercial cooperation to help us to proofread the revision", there are still a large number of mistakes, not just on the grammar but also on the correct use of scientific terms, which hinders understanding. The revised manuscript includes new data to the already large amount of data presented in the original submission. However, the key experiments that would support their model, presented in Figure 9 and constituting the title of the manuscript, are missing.

The authors proposed that in the absence of stress signals, the sensor VgrS autophosphorylates, and then serves as phosphodonor to its partner, VgrR, a DNA binding protein that in its phosphorylated state would not be able to bind to the promoter regions of the genes it activates. But when the bacterium experiences osmotic stress, the protease Prc cleaves VgrS, which prevents its ability to autophosphorylate, resulting in unphosphorylated VgrR, which binds to its promoter sequences resulting in transcription of genes protecting the organism from osmotic stress. This model makes several key predictions, which are readily testable, but not provided in the manuscript.

First, purified phosphorylated VgrR should not bind (or bind less well) to the promoters bound by unphosphorylated VgrR.

Second, there should be less phosphorylated VgrR following osmostress than under non-stress conditions in vivo. This can be examined using Phos-tag gels.

Third, phosphorylated VgrS can serve as phosphodonor to VgrR in vitro.

Four, does VgrS exhibit phosphatase activity towards VgrR-P? If so, is this activity different between full-length and Prc-cleaved VgrS?

Can the authors rule out that the sensing of osmotic stress is actually done by Prc, as opposed to VgrS? What if Prc is activated under osmotic stress so that it cleaves VgrS and VgrS senses something else?

Finally, the manuscript contains a large number of incorrect statements and/or interpretation of data. Because of their large number, I will only provide a few examples. For instance, in the abstract, the authors state that "histidine kinaes are extracytoplasmic receptors" when the best studied HK, CheA, is a cytoplasmic protein. Although CheA is a HK, it relies on other proteins to do the sensing; yet, even for "classical" sensors such as the phosphate sensor PhoB, phosphate sensing occurs in the cytoplasm (e.g., Genes Dev 32:79). Moreover, the authors state that VgrS is a sensor of ferrous iron in page 9 but of ferric iron in page 22. Furthermore, contrary to what the authors write in page 15, second paragraph, the vgrRD51A mutant behaves like the WT or complemented strain, which is in contrast to the behavior of the vgrSH186A and the vgrS strain.

Reviewer #2 (Remarks to the Author):

Thank you very much for taking this reviewers comments seriously and providing such a thorough revision/rebuttal. My comments have satisfactorily been taken care of and do not have any additional criticism.

Reviewer #3 (Remarks to the Author):

The first round of review was influenced by several points of confusion, mainly the size of the cleavage and the virulence data, which were easy to spot. The revised manuscript is substantially reorganized. In thoroughly reading the revision, the reviewer identified some new questions. Most importantly is the rationale behind the use of the delta9 mutant version of VgrS in an otherwise WT background (Fig. 6 & Supplementary Fig. 7 in the new manuscript). This strategy seems flawed for two related reasons: first, the mutant mimics the product after Prc cleavage. Under osmotic stress, VgrSdelta9 is what the WT VgrS becomes. What is the rationale of expressing the product (VgrSdelta9) when both the substrate (full length VgrS) and enzyme (Prc) are functioning normally? Second, VgrSdelta9 is a non-functional protein (i.e. a dead protein), as the author's model shows. It is the product of a reaction meant to shut down the histidine kinase. The transformed line therefore should be indistinguishable from WT. How would the authors explain any phenotype caused by a non-functional protein? It would be much more informative to transform the uncleavable version (used in Fig. 5g), a constitutive kinase expected to have a dominant effect.

My original comment 3 was not really addressed: the authors tried to explain the difficulty with detecting a 1 kDa difference in the endogenous 43 kDa VgrS protein, but I was referring to two recombinant proteins used in their figures. My question actually contains two points: In the revised Fig. 4C, the intact VgrS sensor domain is no more than 14 kDa. Cleavage by Prc should produce a ~13 kDa product. One should be able to resolve this difference. The second point is about the HA-tagged VgrS in the new Fig. 4E. Because of the added HA tag, I calculated the cleavage of this protein should remove 12 kDa from the protein. The anti-VgrS antibody should be able to detect it. The authors' explanation does not apply to these two situations.

Point-to-Point Response to Comments

Reviewers' comments:

Reviewer #1 (Remarks to the Author):

The revised manuscript by Deng and colleagues provides answers to many of the questions that were raised by the three reviewers of the original submission. Although the authors indicate that they “invited a commercial cooperation to help us to proofread the revision”, there are still a large number of mistakes, not just on the grammar but also on the correct use of scientific terms, which hinders understanding. The revised manuscript includes new data to the already large amount of data presented in the original submission. However, the key experiments that would support their model, presented in Figure 9 and constituting the title of the manuscript, are missing.

Response 1-1: We thank the referee to highlight the problems in the language and scientific description. We carefully modified the manuscript to make it clearer. A number of terms or descriptions were changed to avoid mistakes or confusion. After revision, we invited another commercial cooperation to help us proof-read the manuscript (by Lesley Benyon, PhD, from Edanz Group China). The corresponding modifications in the revision were shown in red. We hope that these modifications could enhance the quality of the manuscript.

According to the comments, we designed and performed a number of experiments to solve the mentioned questions associated with our model. Please refer to the following responses for details.

The authors proposed that in the absence of stress signals, the sensor VgrS autophosphorylates, and then serves as phosphodonor to its partner, VgrR, a DNA binding protein that in its phosphorylated state would not be able to bind to the promoter regions of the genes it activates. But when the bacterium experiences osmotic stress, the protease Prc cleaves VgrS, which prevents its ability to autophosphorylate, resulting in unphosphorylated VgrR, which binds to its promoter sequences resulting in transcription of genes protecting the organism from osmotic stress. This model makes several key predictions, which are readily testable, but not provided in the manuscript.

First, purified phosphorylated VgrR should not bind (or bind less well) to the promoters

bound by unphosphorylated VgrR.

Second, there should be less phosphorylated VgrR following osmostress than under non-stress conditions *in vivo*. This can be examined using Phos-tag gels.

Third, phosphorylated VgrS can serve as phosphodonor to VgrR *in vitro*.

Four, does VgrS exhibit phosphatase activity towards VgrR-P? If so, is this activity different between full-length and Prc-cleaved VgrS?

Response 1-2: We appreciate the suggestion to improve the work. Depending on these comments, we conducted new experiment to answer the questions.

The first question: Yes, according to the molecular model provided by Fig. 9, the affinity of the unphosphorylated VgrR in binding the gene promoter should be higher than that of the phosphorylated VgrR (VgrR-P). This experimental evidence is lacking in the previous version. To challenge this hypothesis, the DNA probes of five downstream gene promoters (PXC0711, PXC0943, PXC2164, PXC3300 and PXC3301) were labelled by 5'-FAM (carboxyfluorescein), and then microscale thermophoresis assay (MST) was used to quantify the binding affinity between the DNA probe and VgrR/VgrR-P. As shown in Supplementary Fig. 9, when the VgrR was phosphorylated (VgrR-P) by the histidine kinase VgrS, the dissociation constant (K_d) of the DNA-[VgrR-P] interactions were significantly increased or cannot be determined (Supplementary Fig. 9, middle panels), strongly suggesting that the VgrR-P did not bind these DNA probes or the binding affinity was remarkably decreased. As for negative controls, when the inactive VgrS (VgrS^{H186A}) that cannot be autophosphorylated was added in the reaction, the binding affinity of DNA-VgrR interactions were similar to those of the dephosphorylated VgrR (Supplementary Fig. 9, left and right panels). Collectively, these evidences demonstrate that the phosphorylated VgrR (VgrR-P) cannot bind the promoter regions of the regulated genes or the binding affinity is significantly decreased. The result was added as Supplementary Fig. 9 and we added a paragraph in the maintext accordingly, please refer to Lines 372–383 in revision.

Fig. R1. Detection of phosphorylated VgrS and VgrR by Phos-tag gel. VgrS and VgrR were detected by western blotting using anti-His₆ antibody. **Left panel:** Purified VgrS-His₆ was phosphorylated by ATP, and then VgrS-His₆ and VgrR-His₆ proteins were separated by Phos-tag gel and detected by western blotting. **Right panel:** VgrR were detected in the recombinant strain of ΔvgrR-vgrR-his₆. ΔvgrR-vgrR strain was used as negative control, and purified VgrR-His₆ protein was used as positive control. Note that signals of phosphorylated VgrS and VgrR were not detected *in vitro* or *in vivo*.

The second question: In the study area of TCS, it is technically hard to detect the

phosphorylated HKs or RRs *in vivo* since the signal is weak or their half-lives are very short. Consequently, isotope-based radio-autography has to be used to study the enzymatic property of HK-RR phosphorylation *in vitro* (Scharf, 2010. *Current Opin Microbiol*, 13:246-252). Unfortunately, after repeated efforts, we found that VgrS and VgrR are not exceptions. Phosphorylation of VgrS-VgrR can be observed *in vitro* by isotope-labelling (Fig. 4a–4c in maintext), but Phos-tag gel failed to detect the phosphorylation bands of VgrS and VgrR even the purified proteins were used in the phosphorylation reaction (Fig. R1 in this response, left panel). In recombinant strain, although the band representing VgrR can be detected by western blotting, there is no additional band detected (Fig. R1 in this response, right panel). Currently the *in vivo* phosphorylation level of VgrS or VgrR is technically hard to be measured. However, genetic evidences have suggested that dephosphorylation of VgrS or VgrR promotes the bacterial resistance to osmotic stress since *vgrS*^{H186A}, *vgrS*^{A9g-Q10A} and *vgrR*^{D51A} grew faster than the WT strain under stress (Fig. 6 in maintext). In addition, the dephosphorylated VgrR binds promoter regions of downstream genes with higher affinity than the phosphorylated VgrR, as shown in the afore-mentioned Supplementary Fig. 9.

The third question: Yes, in our previous study, we have reported that the recombinant VgrS can be autophosphorylated and then transfer the phosphoryl group onto VgrR *in vitro* (Fig. 2B and S2 Fig. in Wang et al., 2016. *PLoS Pathogens*, 12:e1006133). According to this comment and previous report, we designed a new experiment to observe the effect of Prc proteolysis on this biochemical process. As shown in the following Fig. R2, the addition of Prc, rather than its inactive form (Prc^{S475A}), not only decreased the autophosphorylation level of VgrS, but also decreased the phosphorylation level of VgrR (Fig. R2, lane 6 and lane 8). The result suggests that Prc proteolysis negatively regulates the phosphorylation of VgrS-VgrR TCS. We added this new result as Fig. 4c in the maintext and describe it in Lines 208–214.

Fig. R2. Prc proteolysis inhibits the phosphorylation of VgrS and VgrR. **Upper panel.** Full length VgrS embedded in the inverted membrane vesicles (IMV, 10 μ M) was phosphorylated by [γ -³²P]ATP. Before addition of 10 μ Ci [γ -³²P]ATP, active Prc or inactive Prc^{S475A} (2 μ M) was added into the mixture, respectively. 10 μ M of VgrR was added into the mixture if needed. VgrS^{H186A} IMV was used as negative control of autophosphorylation. **Lower panel:** Coomassie bright blue-stained gel, used for check the amount of loaded protein.

The fourth question. Thanks for the suggestion. Yes, VgrS potentially has a phosphatase activity to dephosphorylate VgrR and it is interesting to investigate the effect of Prc proteolysis on the phosphatase activity. However, the phosphatase activity of full-length VgrS

towards VgrS is really hard to be studied because the half-life of the phosphorylated VgrR is just around 10 seconds: As the above Fig. R2 and our previous study reported (Fig. 2B and 7B in Wang et al., 2016. *PLoS Pathogens*, 12:e1006133), the signal of phosphorylated VgrR cannot be detected too early (5 second) or too late (> 30 second). In addition, we obtained ³²P-labelled acetyl phosphate that was catalyzed by an acetate kinase. This isotope-labelled small chemical can phosphorylate VgrR, but the half-life of VgrR-P was also near 10 sec. For this reason, we cannot obtain a stable VgrR-P as a substrate to measure the phosphatase activity of the full-length VgrS because it is not easy to complete desalination, remove of VgrS-P membrane, addition of ADP and new VgrS membrane in such a short time window.

Can the authors rule out that the sensing of osmotic stress is actually done by Prc, as opposed to VgrS? What if Prc is activated under osmotic stress so that it cleaves VgrS and VgrS senses something else?

Response 1-3: According to this suggestion, we designed an experiment to determine if the osmotic stress directly activates the Prc protease activity. As shown in the following Fig. R3, addition of various concentrations (20 to 1,000 μ M) of sorbitol in the reaction did not increase the Prc activity in degrading protein substrate β -casein. Therefore, it is unlikely that Prc acts as a sensor of osmotic stress. As previous studies reported, the Tsp/Prc family proteases usually recruit adaptor proteins to recognize the substrates or activate these proteases (Kuhlmann & Chien. 2017. *Curr Opin Microbiol*, 36:118-127). Currently we did not identify the adaptor of Prc in cleaving VgrS sensor. Therefore, we added a description in the Discussion to address this point. Please refer to Lines 471–480 in the revision.

Fig. R3. Osmotic stress did not directly stimulate the protease activity of Prc in degrading β -casein. Various concentrations of sorbitol from 20 to 1,000 μ M were added in the reaction mixture.

*Finally, the manuscript contains a large number of incorrect statements and/or interpretation of data. Because of their large number, I will only provide a few examples. For instance, in the abstract, the authors state that “histidine kinases are extracytoplasmic receptors” when the best studied HK, CheA, is a cytoplasmic protein. Although CheA is a HK, it relies on other proteins to do the sensing; yet, even for “classical” sensors such as the phosphate sensor PhoB, phosphate sensing occurs in the cytoplasm (e.g., *Genes Dev* 32:79). Moreover, the authors state that VgrS is a sensor of ferrous iron in page 9 but of ferric iron in page 22. Furthermore, contrary to what the authors write in page 15, second paragraph, the *vgrR*^{D51A} mutant behaves like the WT or complemented strain, which is in contrast to the behavior of*

the vgrS^{H186A} and the vgrS strain.

Response 1-4: We thank the referee to point out these problems. We carefully checked the manuscript and revised the descriptions to avoid mistakes and confusions, including the afore-mentioned descriptions. A commercial cooperation (Edanz, China) was invited to help us to edit the language of the revision.

1. Yes, as the comment indicated, a portion of HKs did not have transmembrane helices and act as sensors to detect intracellular stimuli. We have revised this sentence, please refer to Lines 397–398 in the revision.

2. Our previous study demonstrated that VgrS specifically binds ferric iron, rather than ferrous iron. There is a mistake in the sentence. We have revised it throughout the manuscript.

3. The vgrR^{D51A} mutant was constructed by *in trans* providing a recombinant pHM1::vgrR^{D51A} in the background of vgrR null mutant. Under osmostress condition, it grew faster than the WT strain during the 0-64 hours (Fig. 6f). To avoid inconsistency, we revised the description in the P15. Please refer to Lines 298–302 for detail.

Reviewer #2 (Remarks to the Author):

Thank you very much for taking this reviewers' comments seriously and providing such a thorough revision/rebuttal. My comments have satisfactorily been taken care of and do not have any additional criticism.

Response 2-1: We thank the reviewer for the valuable comments and positive evaluation of the revision.

Reviewer #3 (Remarks to the Author):

The first round of review was influenced by several points of confusion, mainly the size of the cleavage and the virulence data, which were easy to spot. The revised manuscript is substantially reorganized.

In thoroughly reading the revision, the reviewer identified some new questions. Most importantly is the rationale behind the use of the $\Delta 9$ mutant version of VgrS in an otherwise WT background (Fig. 6 & Supplementary Fig. 7 in the new manuscript). This strategy seems flawed for two related reasons: first, the mutant mimics the product after Prc cleavage. Under osmotic stress, vgrS ^{$\Delta 9$} is what the WT VgrS becomes. What is the rationale of expressing the product (VgrS ^{$\Delta 9$}) when both the substrate (full length VgrS) and enzyme (Prc) are functioning normally? Second, vgrS ^{$\Delta 9$} is a non-functional protein (i.e. a dead protein), as the author's model shows. It is the product of a reaction meant to shut down the histidine kinase. The transformed line therefore should be indistinguishable from WT. How would the authors explain any phenotype caused by a non-functional protein? It would be much more informative to transform the uncleavable version (used in Fig. 5g), a constitutive kinase expected to have a dominant effect.

Response 3-1: We thank this reviewer to offer further comments and suggestion on the

biological function of VgrS^{Δ9}, which help us to improve the work.

In the experiment of Fig. 6, the vgrS^{Δ9} mutant is not constructed in the genetic background of WT by providing a recombinant pHM1:: vgrS^{Δ9} vector, it is an in-frame deletion mutant with the 2nd-9th genetic codes of the vgrS being removed from the bacterial chromosome. Therefore, there is no full-length vgrS copy in this mutant to interfere with the regulatory function of the truncated vgrS^{Δ9}. We thought that the confusion was caused by the previous description of the mutant and that we have revised the corresponding descriptions to make it clearer. Please refer to Lines 286–288 for detail.

In this vgrS^{Δ9} mutant, since the phosphorylation level of VgrS and the cognate RR VgrR were decreased, the mutant grew even faster than the WT strain under osmotic stress condition, while genetic complementation by a full-length vgrS decreased the growth rate to the level of the WT strain. In addition, as shown in Fig. 6h, growth of the Δprc mutant was almost completely arrested under osmotic stress, however, in the double mutant of Δprc-vgrS^{Δ9} or Δprc-ΔvgrS, the vgrS mutations effectively suppressed the growth deficiency caused by prc deletion. This result of epistatic analysis demonstrates that functional inactivation of VgrS autokinase, either by removing of it N-terminal sequence of VgrS, or by deletion of the gene, is important for bacterial survival in the stress condition. The corresponding revision is in the Lines 304–311.

Yes, previous Fig. 5d revealed that substitution of the cleaving site of VgrS resulted in resistance to Prc proteolysis. Accordingly, we have constructed a recombinant strain by substituting the cleaving site of the VgrS (vgrS^{A9G-Q10A}). As shown in Fig. 6d, under the osmotic stress, the growth of this mutant was the slowest, even than the WT strain and ΔvgrS-vgrS strain. These genetic and biochemical results suggest that constitutive phosphorylation of VgrS is detrimental to the osmotic stress resistance of the bacterium, which is in accordance with the prediction. The modification is in Lines 293–293.

My original comment 3 was not really addressed: the authors tried to explain the difficulty with detecting a 1 kDa difference in the endogenous 43 kDa VgrS protein, but I was referring to two recombinant proteins used in their figures. My question actually contains two points: In the revised Fig. 4C, the intact VgrS sensor domain is no more than 14 kDa. Cleavage by Prc should produce a ~13 kDa product. One should be able to resolve this difference. The second point is about the HA-tagged VgrS in the new Fig. 4E. Because of the added HA tag, I calculated the cleavage of this protein should remove 12 kDa from the protein. The anti-VgrS antibody should be able to detect it. The authors' explanation does not apply to these two situations.

Response 3-2: We thank the referee to give us more suggestion on the results of Prc cleavage. In the revised Fig. 4c, the Prc treatment resulted in the cleavage of VgrS sensor. Between the co-incubation of 60-180 min, two clear bands of VgrS sensor were observed, and then the VgrS sensor was completely degraded after 180 min of treatment, suggesting that Prc cleaves VgrS sensor *in vitro*. Please refer to the Fig. 4c and Lines 216–220 for the revision.

In the experiment of Fig. 4E, we constructed a recombinant strain to detect the Prc cleavage *in vivo*. In this strain, a coding sequence of nine aa-length HA tag (YPYDVPDYA) was inserted into the site between the coding sequences of 6th and 7th aa. of VgrS. Consequently, cleavage of the N-terminal sequence of VgrS by Prc only removed 18 amino

acid residues (MNRYPYDVPDYANIDAFA), with a theoretical molecular weight of 2.14 kDa, rather than 12 kDa. This is a small difference that potentially be detected under appropriate separating condition. Therefore, we continue to optimize the condition of SDS-PAGE electrophoresis by changing the concentration and pH of the PAGE gel. As Fig. 4f in revision shown, in the $\Delta vgrS$ - $vgrS^{HA}$ strain that was challenged by the osmotic stress, western blotting analysis using polyclonal antibody of VgrS revealed an additional band close to the major VgrS bands (Fig. 4f, Lanes 7–10, middle panel), while this additional band is not present in the control strain ($\Delta prc\Delta vgrS$ - $vgrS^{HA}$) whose *prc* gene was deleted (Fig. 4f, Lanes 2–5, middle panel). This result demonstrates that the N-terminal sequence of VgrS is removed by Prc *in vivo*. The description was also modified in the maintext, please refer to Lines 226–232.

REVIEWERS' COMMENTS:

Reviewer #2 (Remarks to the Author):

The second revision of this mns by Deng et al. on proteolysis as a novel regulatory mechanism governing the activity of the sensor kinase VgrS is yet another significant improvement of an already strong manuscript.

The authors can only be thank for taking the comments - that required substantial additional experiments - seriously again and congratulated on the outcome.

As far as I can judge, all comments from the reviewers have been satisfactorily addressed. In the few cases in which the requested experiments did not yield the expected/desired results, e.g. studying the phosphorylation of VgrS by Phos-tag gels, the authors provide convincing arguments/reasons for the failure.

Given the impressive amount of high quality data that support a completely novel mechanism of bacterial signal transduction, this paper is now (more than) ready to be published - at least scientifically.

While the language / quality of writing has improved significantly, it unfortunately still remains an issue, particularly in the newly written sections. Countless sentences start with "To...", the tenses are often used incorrectly and quite a number of awkward phrases/sentences remain. Now that the science is done, I would suggest that the authors contact a professional editing service one last time, e.g. as provided by the publisher of Nature Communications, the SpringerNature group at

https://authorservices.springernature.com/language-editing/?utm_source=natureAuthors&utm_medium=referral&utm_campaign=natureAuthor s

Other than that, I am really looking forward to the publication of this scientifically most impressive work.

Point-to-Point Responses to Editorial Requests and Reviewers' Comments

REVIEWERS' COMMENTS:

Reviewer #2 (Remarks to the Author):

The second revision of this mns by Deng et al. on proteolysis as a novel regulatory mechanism governing the activity of the sensor kinase VgrS is yet another significant improvement of an already strong manuscript. The authors can only be thank for taking the comments - that required substantial additional experiments - seriously again and congratulated on the outcome.

As far as I can judge, all comments from the reviewers have been satisfactorily addressed. In the few cases in which the requested experiments did not yield the expected/desired results, e.g. studying the phosphorylation of VgrS by Phos-tag gels, the authors provide convincing arguments/reasons for the failure.

Given the impressive amount of high quality data that support a completely novel mechanism of bacterial signal transduction, this paper is now (more than) ready to be published - at least scientifically.

Response 1. We really appreciate the reviewer to evaluate our manuscript for the third time and give us positive judgement. These comments, together with those provided by the other two referees, remarkably promote the quality of the work. We learned a lot from this valuable suggestion.

While the language / quality of writing has improved significantly, it unfortunately still remains an issue, particularly in the newly written sections. Countless sentences start with "To...", the tenses are often used incorrectly and quite a number of awkward phrases/sentences remain. Now that the science is done, I would suggest that the authors contact a professional editing service one last time, e.g. as provided by the publisher of Nature Communications, the SpringerNature group at https://authorservices.springernature.com/language-editing/?utm_source=natureAuthors&utm_medium=referral&utm_campaign=natureAuthors

Other than that, I am really looking forward to the publication of this scientifically most impressive work.

Response 2. Thank you very much to point out the problem and carefully revised the language in the maintext. According to the suggestion and the Editorial Requirement, we

modified the manuscript. In addition, we invited an expert of the Springer Nature Author Service to help us editing the revision. All the modifications were tracked in the Word file for in-depth review. We hope that these efforts improve the quality of the language of it.